# EGonc : Energy-based Open-Set Node Classification with substitute Unknowns

**Qin Zhang**[1]  **Zelin Shi**[1]  **Shirui Pan**[2]  **Junyang Chen**[1]  **Huisi Wu**[1]  **Xiaojun Chen**[1]*

[1]Shenzhen University  [2]Griffith University

qinzhang@szu.edu.cn, shizelin2021@email.szu.edu.cn,
s.pan@griffith.edu.cn, {junyangchen,hswu,xjchen}@szu.edu.cn

## Abstract

Open-set Classification (OSC) is a critical requirement for safely deploying machine learning models in the open world, which aims to classify samples from known classes and reject samples from out-of-distribution (OOD). Existing methods exploit the feature space of trained network and attempt at estimating the uncertainty in the predictions. However, softmax-based neural networks are found to be overly confident in their predictions even on data they have never seen before and the immense diversity of the OOD examples also makes such methods fragile. To this end, we follow the idea of estimating the underlying density of the training data to decide whether a given input is close to the in-distribution (IND) data and adopt Energy-based models (EBMs) as density estimators. A novel energy-based generative open-set node classification method, *EGonc*, is proposed to achieve open-set graph learning. Specifically, we generate substitute unknowns to mimic the distribution of real open-set samples firstly, based on the information of graph structures. Then, an additional energy logit representing the virtual OOD class is learned from the residual of the feature against the principal space, and matched with the original logits by a constant scaling. This virtual logit serves as the indicator of OOD-ness. EGonc has nice theoretical properties that guarantee an overall distinguishable margin between the detection scores for IND and OOD samples. Comprehensive experimental evaluations of EGonc also demonstrate its superiority.

## 1 Introduction

Learning on graphs, where instance nodes are inter-connected, has become one of the central problems for deep learning, as relational structures are pervasive and induce data inter-dependence which hinders trivial adaptation of existing approaches that assume inputs to be i.i.d. sampled [56]. However, current models mostly focus on improving testing performance of in-distribution (IND) data and largely ignore the potential risk w.r.t. out-of-distribution (OOD) testing samples that may cause negative outcome if the prediction is overconfident on them [31].

Real-world applications often require machine learning systems to interact with an open world, violating the common assumption that testing and training distributions are identical. This urges the community to devote increasing efforts on how to enhance models' generalization [9] and reliability [47] *w.r.t.* out-of-distribution data, by classifying samples from known classes and rejecting samples from out-of-distribution.

Existing methods in addressing this task normally are achieved by assuming that a model is more confident in its predictions for closed-set samples than for open-set samples [55, 8]. Based on this

---

*Corresponding author.

assumption, a threshold is applied to a model's confidence score to separate unknown samples from known ones. Despite its apparent intuitiveness [20], this method often fails because deep neural networks tend to be over-confident in their predictions, assigning high confidence scores even to unknown inputs [19, 14, 1, 42, 71]. Furthermore, determining an optimal threshold for distinguishing known from unknown classes is a challenging and time-consuming task [38, 12, 22]. Others exploit the feature space of trained network and attempt at estimating the density of in-distribution (IND) features to address OOD detection [29], such as Gaussian Mixture Models [31, 44], nearest neighbors distribution [49] and energy logits [34, 8]. However, the immense diversity of the OOD examples makes such methods fragile [52]. For example, GMMs' density explicitly decreases when moving away from training data, making them effective for far-OOD detection, while energy logits benefits from the classifier training to obtain strong results on near-OOD samples [29, 52].

To overcome these limitations, researchers try to make advantages of unrelated samples in the open-set environment [21]. Nevertheless, the methods using real open-set samples to help the training of models often require a substantial amount of data [23, 48, 5], and insouciantly selected data does not really help [66]. Well-designed outlier samples are difficult to obtain and require experts for identification and labeling [6, 25].

In this paper, we follow the idea of estimating the underlying density of the training data to decide whether a given input is close to the IND data and adopt Energy-based models (EBMs) as density estimators. To further improve the OOD detection performance, we try to generate substitute unknown samples carefully under the guidance of graph structure, to mimic the distribution of real open-set classes. Specifically, a novel energy-based generative open-set node classification method, *EGonc*, is proposed to achieve open-set graph learning. Based on the generated substitute unknowns, an additional energy logit representing the virtual OOD class is learned from the residual of the feature against the principal space, and matched with the original logits by a constant scaling. This virtual logit serves as the indicator of OOD-ness. EGonc has nice theoretical properties that guarantee an overall distinguishable margin between the detection scores for IND and OOD samples. Comprehensive experimental evaluations of EGonc also demonstrate its superiority. To sum up, the contributions of this paper are summarized as follows:

- A novel method, EGonc, for open-set node classification is proposed by redefining the open-world graph learning paradigm based on the energy model and elaborate unknown-substitute generation;
- EGonc has nice theoretical properties that guarantee an overall distinguishable margin between the detection scores for IND and OOD samples. The adopted energy regularization loss has consistent effects with the cross-entropy loss as well as with the tailored complement entropy loss on the known classes, and that they are not mutually exclusive;
- No open-set data (samples of unknown classes or any side information of unknown classes) is required during training and validation. EGonc has an explicit classifier for unknowns and does not require threshold tuning;
- EGonc is agnostic to specific GNN architecture and demonstrates robust generalization capabilities.
- Experiments conducted on benchmark graph datasets illustrate the commendable performance achieved by EGonc.

## 2 Background

### 2.1 Open-set Node Classification

This paper focuses on the node classification problem for a graph. Consider a *graph* denoted as $G = (V, E, X)$, where $V = \{v_i | i = 1, \ldots, N\}$ is a set of $N$ nodes in the graph. $E = \{e_{i,j} | i, j = 1, \ldots, N. \ i \neq j\}$ is a set of edges between pairs of nodes $v_i$ and $v_j$. $X \in \mathbb{R}^{N \times d}$ denotes the feature matrix of nodes, and $d$ is the dimension of node features. $x_i \in X$ indicates the feature vector associated with each node $v_i$. The topological structure of $G$ is represented as an adjacency matrix $A \in \mathbb{R}^{N \times N}$, where $A_{i,j} = 1$ if the nodes $v_i$ and $v_j$ are connected, *i.e.*, $\exists e_{i,j} \in E$, otherwise $A_{i,j} = 0$. $Y \in \mathbb{R}^{N \times C}$ is the label matrix of $G$, where $C$ is the already-known node classes. If node $v_i \in V$ is associated with a label $c$, $y_{i,c} = 1$, otherwise, $y_{i,c} = 0$.

For a typical *closed-set node classification* problem, a GNN encoder $f_{\theta_g}$ takes node features $X$ and adjacency matrix $A$ as input, aggregates the neighborhood information and outputs representations. Then, a classifier $f_{\theta_c}$ is used to classify the nodes into $C$ already-known classes. The GNN encoder

and the classifier are optimized to minimize the expected risk [63] in Eq. (1), with the assumption that test data $\mathcal{D}_{\text{te}}$ and train data $\mathcal{D}_{\text{tr}}$ share the same feature space and label space, *i.e.*,

$$f^* = \arg\min_{f \in \mathcal{H}} \mathbb{E}_{(x,y) \sim \mathcal{D}_{\text{te}}} \mathbb{I}(y \neq f(\theta_g, \theta_c; x, A)) \tag{1}$$

where $\mathcal{H}$ is the hypothesis space, $\mathbb{I}(\cdot)$ is the indicator function which outputs 1 if the expression holds and 0 otherwise. Generally, it can be optimized with cross-entropy to discriminate between known classes.

In **open-set node classification problem**, given a graph $G = (V, E, X)$, $\mathcal{D}_{\text{tr}} = (X, Y)$ denotes the train nodes. $\overline{\mathcal{D}}_{\text{te}} = (X_{\text{te}}, Y_{\text{te}})$ is the test nodes, where $X_{\text{te}} = S \cup U$, $Y_{\text{te}} = \{1, \ldots, C, C+1, \ldots\}$. $S$ is the set of nodes that belong to seen classes that already appeared in $\mathcal{D}_{\text{tr}}$ and $U$ is the set of nodes that do not belong to any seen class (*i.e.*, unknown class nodes). Open-set node classification aims to learn a $(C+1)$-*class* classifier $f_{\overline{\theta}_c}$ that $f(\theta_g, \overline{\theta}_c; X_{\text{te}}, \overline{A}) : \{X_{\text{te}}, \overline{A}\} \rightarrow \overline{\mathcal{Y}}$, $\overline{\mathcal{Y}} = \{1, \ldots, C, unknown\}$, with the minimization of the expected risk [63]:

$$\overline{f}^* = \arg\min_{f \in \mathcal{H}} \mathbb{E}_{(x,y) \sim \overline{\mathcal{D}}_{te}} \mathbb{I}(y \neq f(\theta_g, \overline{\theta}_c; x, \overline{A})) \tag{2}$$

where $\overline{A}$ is the adjacency matrix for $X_{\text{te}}$. The predicted class $unknown \in \overline{\mathcal{Y}}$ contains a group of novel categories, which may contain more than one class. Thus, the overall risk aims to classify known classes while also detecting the samples from unseen categories as class $unknown$.

It is worth mentioning that *open-set node classification problem* is different with *out-of-distribution (OOD) detection problem*. The objective of OOD detection is to identify a new sample whether from unknown classes or not, which is a binary classification challenge. Thereby, OOD detection techniques are not directly applicable to open-set node classification scenarios. This emphasizes the crucial importance of conducting research on open-set node classification, highlighting the distinct requirements and challenges within this domain.

**From close-set to open-set classifier**, an intuitive way is thresholding [20]. Taking the max output probability as confidence score, i.e., $conf = max_{c=1,\ldots,C} f_c(x, A)$, it assumes the model is more confident in closed-set instances than open-set ones. Then we can extend a closed-set classifier by

$$\hat{y}_i = \begin{cases} \arg\max_{c=1,\ldots C} f_c(x_i, A), & conf > \tau \\ unknown, & otherwise. \end{cases} \tag{3}$$

where $\tau$ is a threshold. However, due to the overconfidence phenomena of deep neural networks, the output confidence of known and unknown is both high [1]. As a result, tuning a threshold that well separates known from unknown is hard and time-consuming.

Differently, the approach presented in this paper redefines the paradigm of open-set graph learning from the perspective of energy models. well-designed generated unknown substitutes are introduced to aid in training of the energy model performance and elaborate loss function to improve classification performance. There are advantages of adding a class instead of optimizing a threshold as in previous methods [46, 54, 1], since it does not require real open-set samples in validation and also eliminates the difficulty of tuning the threshold.

## 2.2 Energy-based Models

Energy-based models [30] use the energy-function $E_\theta$ which defines a density over the data $x$ as $p_\theta = \frac{exp(-E_\theta(x))}{Z(\theta)}$, where $\theta$ are learnable parameters and $Z(\theta) = \int exp(-E_\theta) dx$ is the normalizing constant of the EBM. $Z_\theta$ ensures that the induced density function Equation integrates to 1. However, $Z_\theta$ is often hard to compute or even approximate since it is a high-dimensional integral. On the positive side, $E_\theta$ can be any function $E_\theta : \mathbb{R}^D \rightarrow \mathbb{R}$ placing no restrictions on the transformations compared to Normalizing Flows [68].

We can directly use the energy for OOD detection. That is, since a data point having high probability is equivalent to having low energy as $p_\theta(x) \propto -E_\theta(x)$. This means we are not required to estimate the normalizing constant $Z(\theta)$ in practice by considering a decision threshold $\tau$ and binary classifier $G$ for OOD data as

$$G(x, \tau, \theta) = \begin{cases} 0, & \text{if } -E_\theta(x) \leq \tau, \\ 1, & \text{if } -E_\theta(x) > \tau \end{cases} \tag{4}$$

where 1 corresponds to IND samples and 0 to OOD samples.

We can additionally employ EBMs to $C$-category classification problem as discriminative EBMs [13]. Suppose $f_\theta : \mathbb{R}^D \to \mathbb{R}^C$ is a classifier assigning logits for $C$ classes for a data point $x \in \mathbb{R}^D$. The logits can be interpreted as unnormalized probabilities of the joint distribution $p_\theta(x, y) = \frac{exp(f(x)[y])}{Z(\theta)}$, yielding the marginal distribution over $x$ as $p_\theta(x) = \sum_y p_\theta(x, y)$. For training, the factorization can be used, $i.e.,\log p_\theta(x, y) = \log p_\theta(x) + \log p_\theta(y|x)$, and a standard Cross Entropy loss [2] is usually employed to optimize $p_\theta(y|x)$.

To sum up, the connection between Energy-Based Model and a discriminative neural classifier is established by defining the energy function as the negation of the predicted logit value: $E(\mathbf{x}, y) = -h(\mathbf{x})_{[y]}$. Moreover, the energy function $E(x)$ can be computed for any given input

$$E(\mathbf{x}) = -\log \sum_{y'} e^{-E(\mathbf{x}, y')} \tag{5}$$

Nevertheless, to the best of our knowledge, existing research on energy-based modeling as well as its application is mostly concentrated on OOD detection, where the primary emphasis is on $i.i.d.$ inputs while the power of modeling inter-dependent data has remained further exploration. Besides, there have been relatively few attempts to apply energy-based models in open-world scenarios, and this is an endeavor we will also pursue.

## 3 Methodology

We propose a unified open-set node classification framework based on energy scores and generated substitute unknowns. The energy scores mitigate a critical problem of softmax confidence with arbitrarily high values for OOD examples [18] and the differences of energies between in-distribution and out-of-distribution allow effective differentiation.

Specifically, a GNN encoder $f_{\theta_1}$ takes node features $X$ and adjacency matrix $A$ as input, aggregates the neighborhood information and outputs representation $h_i^k$ for each node $v_i$ in its $k$-th layer, $k = 1, \ldots, \mathcal{K}$, and $\mathcal{K}$ is the total number of layers. Thus, in the $k$-th layer, for node $v_i$, the GNN encoder aggregates neighbor information from the $k$-1 layer into a neighborhood representation:

$$h_i^k = f^k(\theta_1; h_i^{k-1}, h_j^{k-1}, j \in \mathcal{N}_i) \tag{6}$$

where $h_i^k \in \mathbb{R}^{d_k}$ is the hidden representation at the $k$-th layer, $\mathcal{N}_i$ is the neighborhood of node $v_i$ and $h^0 = X$.

Meanwhile, we mimic novel patterns by generating substitute unknown nodes through manifold mixup [51, 17]. Targeted node pairs are selected and mixed up at the middle layer of the GNN, and two kinds of substitutes are created: inter-class unknown substitutes that separate known classes from each other and external unknown substitutes that separate known from unknown. Generated substitutes and original known class nodes are input into the remaining layers together to learn discriminative representations.

Then, a classifier $f_{\theta_2}$ is learned to classify the nodes into $C + 1$ classes according to their energy scores. The energy scores of node $v_i$ is obtained through energy propagation upon the graph topology, $i.e.,$

$$E_i^k = f^k(E_i^{k-1}, E_j^{k-1}, j \in \mathcal{N}_i) \tag{7}$$

where $E_i^0 = -\log \sum_{y'} e^{f_{\theta_2}(h_i^{\mathcal{K}})[y']}$ is the initial energy score of $v_i$.

To conclude, the proposed EGonc method consists of three main modules: 1) a *substitute node generation* module to create substitute unknown class nodes; 2) a *energy propagation* module to obtain energy score of each node; and 3) an *open-set classifier learning* module to guarantee the classification of known classes and the rejection of the unknown class. We introduce them in details.

### 3.1 Substitute Unknown Generation

As mentioned previously, insouciantly selected unrelated data does not really help [66] the open-set classification. Thus, we try to generate substitute unknown nodes strategically positioned at class

boundaries. These nodes play a crucial role in the learning process by providing supervision and distinguishing between known and unknown nodes. Specifically, we generate two types of substitutes, inter-class unknown substitutes and external unknown substitutes, through *manifold mixup* [51, 17].

For a well-trained classifier, nodes with common categories tend to have similar characteristics, while those from different classes often have distinct ones. Ideally, well-defined boundaries separate the different classes. Considering that nodes close to these boundaries may have less representative features for their own classes, we chose to generate substitute unknowns using nodes close to these boundaries regions. As a result, samples belonging to different classes, but located in close proximity, become optimal candidates for substitute generation. *i.e.*, pairs of nodes in distinct classes connected by an edge. To be specifical, given two connected nodes from different categories, denoted as $\{(x_i, y_i), (x_j, y_j)\}$, where $y_i \neq y_j$ and $a_{ij} = 1$, their embeddings in the $k$-th layer are $h_i^k$ and $h_j^k$, derived by inputting the graph into the network $f_{\theta_1}^{1 \sim k}$. The inter-class unknown substitute $(\tilde{x}_i, \tilde{y}_i)$ is obtained as follows:

$$\begin{cases} \tilde{x}_i = \alpha h_i^k(\theta_1; x_i, A) + (1 - \alpha) h_j^k(\theta_1; x_j, A) \\ \tilde{y}_i = C + 1. \end{cases} \tag{8}$$

Where $\alpha \in [0, 1]$ is randomly sampled from a Beta distribution, typically centred around 0.5. $\tilde{y}_i = C + 1$ denotes the categories index of $\tilde{x}_i$, indicating it belongs to the *unknown* class. Two edges between $(x_i, \tilde{x}_i)$ and $(x_j, \tilde{x}_i)$ are also introduced into the graph. We refer to the set of generated inter-class samples as $X_{\text{int}}$. The purpose of $X_{\text{int}}$ is to simulate the distribution of unknown classes nodes existing between known classes. The mixed hidden representation $\tilde{x}_i$ is then passed to $f_{\theta_1}^{k+1 \sim \mathcal{K}}$ to obtain the final representation $\tilde{h}_i^{\mathcal{K}} = f_{\theta_1}^{\mathcal{K}}(\tilde{x}_i)$.

In addition to the inter-class unknown substitutes, we also generate external unknown substitutes to reflect the unknown distributions at the periphery. We use peripheral nodes of each known category along with their respective class center to generate these external unknown substitutes. The first kind of peripheral nodes are leaf nodes that belong to known classes, denoted as $x_i \in X_{\text{tr}}$, $s.t.$ $\sum_j A_{i,j} = 1$. The second kind are nodes with low classification confidence, namely those nodes with the top $T$ least confident scores within each known classes. In this way, it allows us to identify both structured-based and semantic-based peripheral nodes. We denote the set of peripheral nodes as $X_{\text{per}}$. Then, we obtain the embedding of class centers in the $k$-th layer based on the ground-truth annotations, *i.e.*, $h_{(c)}^k = \frac{1}{|X^c|} \sum_{x_i \in X^c} h_i^k(\theta_1; x_i, A), c = 1, \ldots, C$, where $X^c$ represents the nodes labeled with the class $c$. And we can use the manifold mixup on peripheral nodes and their respective inverted class centers $\{(x_i, y_i), (x_{(y_i)}, y_i), x_i \in X_{\text{per}}, y_i \in \{1, \ldots, C\}\}$ to derive the external unknown substitutes in the $k$-th layer, i.e.,

$$\begin{cases} \tilde{x}_i = \beta h_i^k(\theta_1; x_i, A) + \gamma(-h_{(y_i)}^k) \\ \tilde{y}_i = C + 1. \end{cases} \tag{9}$$

where $\beta > 0$ and $\gamma > 0$ represent two hyperparameters used to adjust the distance and relative position between the generated samples and the existing known class samples. Furthermore, edges connecting $(x_i, \tilde{x}_i), x_i \in X_{\text{per}}$ are also included in the graph. We denote the set of generated external unknown substitutes as $X_{\text{ext}}$ and define unknown substitutes as $X_{\text{sub}} = X_{\text{int}} \cup X_{\text{ext}}$.

## 3.2 Energy Propagation

To further capture the diverse distribution of the unknown classes. we introduced an energy-based model. Through formula $E(\mathbf{x}, y) = -h(\mathbf{x})_{[y]}$, we can establish a bridge between the energy function and an open-set classifier.

**Proposition 1.** The energy score (Eq. (5)) can be an powerful indicator for OOD detection due to its valuable trait: the yielded energy scores for in-distribution data typically exhibit a lower tendency than those of OOD data.

As an energy model on graphs, in order to fully leverage the topological structure information of the graph, we apply energy propagation to the topological structure of the graph.

$$E^{(k)} = \zeta E^{(k-1)} + (1 - \zeta) D^{-1} \hat{A} E^{(k-1)} \tag{10}$$

Here, $\hat{A} = D^{-1/2} A D^{-1/2}$ is the symmetrically normalized adjacency matrix of $A$, and $D$ represents the degree matrix, $E^{(k)} = \left[ E_i^{(k)} \right], x_i \in \mathcal{D}_{\text{tr}} \cup X_{\text{sub}}$. With $0 < \zeta < 1$ as parameter, determine the

energy weights for individual nodes with respect to themselves and other connected nodes. The motivation behind topological energy propagation is to incorporate the underlying physical processes involved in data generation by exploiting local interactions between instances. Since similar nodes are often adjacent in the graph, the model can learn relationships between these nodes through energy propagation. This helps to better capture local structures and similarities within the graph.

**Proposition 2.** For a node $x_i$, if its energy score $E_i^{(k-1)}$ is greater (resp. less) than the average of the energy scores of its one-hop neighbors at the current layer, *i.e.*, $\frac{\sum_j \hat{A}_{ij} E_j^{(k-1)}}{\sum_j \hat{A}_{ij}}$, then the updated energy score of its own yields $E_i^{(k)} < E_i^{(k-1)}$, (resp, $E_i^{(k)} > E_i^{(k-1)}$).

Due to structural homogeneity [60, 28], known classes samples and unknown substitutes often connect within their respective distributions, indicating a correlation. Therefore, by using this energy propagation formula, it could ensure the average energy scores of the known classes samples decrease while the average energy scores of the unknown substitutes increase.

### 3.3 Open-set Classifier Learning

To address the diverse compositions of known and unknown categories, it is imperative to extract invariant information from both known and unknown classes. Consequently, the backbone GNN network $f_{\theta_1}$ is shared across the $C + 1$ classes, facilitating the learning of class distribution and node representations.

To calibrate the closed-set (known classes) classifier to an open-set classifier, an additional category is introduced in the final classification layer to handle unknown predictions. Suppose the weights of the closed-set classifier are denoted as $w_{\text{close}} \in \mathbf{R}^{d_\mathcal{K} \times C}$ where $d_\mathcal{K}$ represents the dimension of the embeddings in the final layer of the GNN, the open-set classifier is formed by combining the closed-set classifier and the substitute classifier, *i.e.*, $\theta_2 = [w_{\text{close}}, w_{\text{sub}}] \in \mathbf{R}^{d_\mathcal{K} \times (C+1)}$, where $w_{\text{sub}} \in \mathbf{R}^{d_\mathcal{K} \times 1}$ represents the weights associated with the substitute classifier. For simplicity, the bias term is omitted in this context. Then, the integrated classifier is trained using both the original known classes samples and the generated substitute samples, *i.e.*, $\overline{\mathcal{D}}_{\text{tr}} = \mathcal{D}_{\text{tr}} \cup (X_{\text{sub}}, Y_{\text{C+1}})$, $X_{\text{sub}=X_{\text{int}} \cup X_{\text{ext}}}$, using the cross entropy loss, *i.e.*,

$$l_1 = \sum_{(x_i,y_i) \in \mathcal{D}_{\text{tr}}} l_{\text{CrE}}(\hat{y}_i, y_i) + \lambda_1 \sum_{x_i \in X_{\text{sub}}} l_{\text{CrE}}(\hat{y}_i, C+1) \tag{11}$$

where $\hat{y}_i = s(f_{\theta_2}(f_{\theta_1}(x_i)))$ is the output vector of the open-set classifier for the input node $x_i$ through softmax $s(\cdot)$. $l_{\text{CrE}}(\hat{y}_i, y_i) = -y_i \log \hat{y}_i$ is the cross entropy loss.

$l_1$ loss primarily uses information from the ground-truth class to maximize the likelihood, it tends to overlook the influence of complementary classes (*i.e.*, incorrect classes) [3]. Therefore, we incorporate the concept of complement entropy [3] to complement the softmax cross entropy, to neutralize the effects of complementary classes.

Specifically, for the generated unknown substitutes, we minimize their inherent complementary loss, *i.e.*, minimize the average of entropy over the $C$ known classes, considering that the generated substitutes may be adjacent to any of the known classes. Conversely, for nodes belong to known classes, they are predominantly adjacent to the generated substitutes, *i.e.*, the $C + 1$ class, we encourage the substitute classifier to assign the second-largest probability to these nodes. In light of these considerations, we introduce a *tailored complement entropy loss* for EGonc as follows:

$$l_2 = \sum_{(x_i,y_i) \in \mathcal{D}_{\text{tr}}} l_{\text{CrE}}(\hat{y}_i \backslash y_i, C+1) + \sum_{x_i \in X_{\text{sub}}} l_{\text{CoE}}(\hat{y}_i, y_i) \tag{12}$$

where $\hat{y}_i \backslash y_i$ denotes the prediction logits after excluding the probability of its corresponding ground-truth label. $l_{CoE} = -\sum_{c=1,c \neq y_i}^{C+1} \frac{\hat{y}_{i,c}}{1-\hat{y}_{i,y_i}} \log \frac{\hat{y}_{i,c}}{1-\hat{y}_{i,y_i}}, \forall (x_i, y_i) \in \overline{D}_{\text{tr}}$. The first term of $l_2$ aims to minimize the misclassification of known class nodes into other known classes. By employing $l_2$ loss, the open-world classifier $f_{\theta_2}$ can accurately classify node belonging to known classes into their respective known class, while leveraging the substitute classifier to establish an appropriate boundary for distinguishing between the known and the unknown.

To mitigate the impact of energy attenuation on unknown substitutes during the energy propagation, we further introduce a strict constraints via energy regularization $l_3$ to the model. Drawing inspiration from the Elastic Network [73, 10], we observe a parallel in structure and purpose between regularization terms and their associated error terms. In light of this observation, we adopt a strategy that incorporates both linear loss and quadratic loss terms to penalize errors effectively. The linear error terms facilitate the generation of sparse solutions, thereby enhancing the model's robustness, while the quadratic error terms are adept at accommodating outliers and achieving a better fit to the data. By integrating these two kinds of error terms, we aim to enhance the adaptability and generalization capacity of the model.

$$l_3 = k_1 \left( \sum_{x_i \in \mathcal{D}_{\text{tr}}} \sigma(E_{\text{ind}}(x_i)) + \sum_{x_j \in X_{\text{sub}}} \sigma(E_{\text{ood}}(x_j)) \right) + k_2 \left( \sum_{x_i \in \mathcal{D}_{\text{tr}}} \sigma(E_{\text{ind}}(x_i))^2 + \sum_{x_j \in X_{\text{sub}}} \sigma(E_{\text{ood}}(x_j))^2 \right)$$
(13)

where $\sigma(\cdot)$ is the LeakyReLU function. $k_1, k_2$ are the weights for linear and quadratic error terms, respectively. $t_{\text{in}}, t_{\text{out}}$ are hyperparameters. $E_{\text{ind}}(x_i) = E(x_i, A; h_\theta) - t_{in}, \forall x_i \in \mathcal{D}_{\text{tr}}$ and $E_{\text{ood}}(x_j) = t_{\text{out}} - E(x_j, A; h_\theta), \forall x_j \in X_{\text{sub}}$ represent the energy error terms correspond to the IND data and the OOD data, respectively. Here, $E(x_i, A; h_\theta)$ represents the energy value of sample $x_i$ at the final layer. In this way, the energy scores learned with the constraint of $l_3$ are of benefit to open-set classification, and it also mitigate the excessive attenuation of energy scores for unknown substitutes caused by energy propagation.

**Proposition 3.** For any energy regularization $l_3$ that ensures that the GNN model $h_{\theta*}$ minimizing $l_3$, where $t_{\text{in}} < t_{\text{out}}$ are two margin parameters, then the corresponding softmax categorical distribution $p(y|x, \mathcal{G}_x; h_{\theta*})$ also minimizes $l_1$, while $p(C + 1|x, \mathcal{G}_x; h_{\theta*})$ similarly minimizes $l_2$. This means $l_3$ is optimized in the same direction as $l_1$ and $l_2$ for ind data.

Finally, the total loss of EGonc is a combination of cross entropy, the tailored complement entropy loss, and energy regularization loss, *i.e.*,

$$l_{\text{total}} = l_1 + \lambda_2 l_2 + \lambda_3 l_3$$
(14)

where $\lambda_2 > 0$ and $\lambda_3 > 0$ are the hyperparameters to balance the losses. The algorithm of EGonc and its complexity analysis are provided in Appendix D.

## 4 Experiments

Experiments were carried out to validate the performance of EGonc. These experiments include: *open-set node classification comparison, ablation study, parameter study*, and *generalization analysis*. Code are available at *https://github.com/hiromisyo/EGonc*.

**Datasets.** Experiments to evaluate the performance for open-set node classification were mainly performed on five benchmark graph datasets [54, 72], namely Cora[2], Citeseer[3], DBLP[4], PubMed[5], and Ogbn_arxiv[6] [24, 45], which are widely used citation network datasets. Statistics are presented in Appendix E.2.

**Metrics.** Accuracy and Macro-F1 are used for performance evaluation.

**Implementation Details.** Generally, EGonc adopt GCN [28] as the backbone neural network for experimental evaluation unless otherwise specified. Detailed model parameters and platform information are given in Appendix E.3.

**Test settings.** Two kinds of open-set classification evaluations were conducted to consider short or long distances between known classes and unknown classes, *i.e.*, near open-set classification and far open-set classification. Specifically, in the near open-set classification experiment, for each dataset, the data of several classes were held out as the unknown classes for testing and the remaining classes

---

[2]https://graphsandnetworks.com/the-cora-dataset/

[3]https://networkrepository.com/citeseer.php

[4]https://dblp.uni-trier.de/xml/

[5]https://pubmed.ncbi.nlm.nih.gov/download/

[6]https://github.com/snap-stanford/ogb/

Table 1: Near open-set classification on five citation network datasets with one unknown class (u=1) in the *inductive learning setting*. Numbers reported are all percentage (%).

| Methods | Cora | | Citeseer | | DBLP | | Pubmed | | Ogbn_arxiv | | Average | |
|---|---|---|---|---|---|---|---|---|---|---|---|---|
| | Acc | F1 | Acc | F1 | Acc | F1 | Acc | F1 | Acc | F1 | Acc | F1 |
| GCN_soft | 70.6 | 67.6 | 44.6 | 38.9 | 63.8 | 59.2 | 28.9 | 29.9 | 49.8 | 17.5 | 51.5 | 42.6 |
| GCN_sig | 69.2 | 64.7 | 45.3 | 44.5 | 63.5 | 58.7 | 28.9 | 29.8 | 48.8 | 9.5 | 51.1 | 41.4 |
| GCN_soft_$\tau$ | 73.6 | 73.8 | 57.3 | 54.5 | 65.0 | 62.4 | 49.7 | 48.6 | 47.3 | 20.6 | 58.6 | 52.0 |
| GCN_sig_$\tau$ | 79.7 | 80.1 | 62.1 | 54.6 | 69.2 | 68.2 | 45.1 | 46.0 | 46.0 | 8.3 | 60.4 | 51.4 |
| Openmax | 74.6 | 75.1 | 56.2 | 54.5 | 67.2 | 67.2 | 49.1 | 48.7 | 45.5 | 16.3 | 58.5 | 52.4 |
| DOC | 77.8 | 78.1 | 66.0 | 56.7 | 69.9 | 69.2 | 45.6 | 46.2 | 46.7 | 20.7 | 61.2 | 52.2 |
| PROSER | 83.2 | 83.7 | 73.7 | 63.6 | 71.7 | 72.6 | 71.0 | 58.4 | 53.0 | 31.1 | 70.5 | 61.9 |
| OpenWGL | 78.1 | 78.9 | 64.1 | 60.8 | 71.4 | 72.2 | 65.3 | 63.4 | 45.4 | 20.7 | 64.9 | 60.2 |
| GNNSAFE | 79.6 | 81.0 | 69.8 | 60.3 | 72.5 | 74.1 | 70.1 | 66.8 | 51.2 | 24.2 | 68.6 | 61.3 |
| $\mathcal{G}^2Pxy$ | 84.3 | 84.8 | 75.5 | 71.0 | 77.3 | 79.0 | 73.7 | 70.2 | 62.7 | **33.0** | 74.7 | 67.6 |
| EGonc | **84.5** | **84.9** | **75.8** | **71.5** | **79.1** | **80.8** | **80.2** | **75.5** | **63.0** | **33.0** | **76.5** | **69.1** |

were considered as the known classes. $70\%$ of the known class nodes were sampled for training, $10\%$ for validation and $20\%$ for testing. In the far open-set classification experiment, nodes from other datasets were used as unknown class samples for testing, other than the nodes from the dataset used in training.

Besides, a comparison is also provided for different settings in terms of the availability of side information on unknown classes during training or validation, known as the inductive learning setting and the transductive learning setting. In experiments with inductive learning setting, there is not any information about the real unknown class (such as the features $x_i$ or side information of unknown classes) used during training or evaluation, while under the transductive learning setting, the whole graph (including sampled known class nodes and unlabeled unknown class nodes) are input during model training or validation.

**Baselines.** We compare EGonc with ten baselines, which can be classified to four categories.

- 1) Closed-set classification methods: *GCN_soft* and *GCN_sig*. They are GCNs [28] with a softmax layer or a multiple 1-vs-rest of sigmoids layer as output layer.

- 2) Open-set classification methods with thresholds: *GCN_soft_$\tau$*, *GCN_sig_$\tau$*, Openmax [1], DOC [46] and OpenWGL [54]. The threshold is chosen from $\{0.1, 0.2, \ldots, 0.9\}$ or as the description of the original paper, to perform open-set recognition.

- 3) Generative Open-set Classification methods: PROSER [70] and $\mathcal{G}^2Pxy$ [67].

- 4) Energy-based methods: GNNSAFE [56].

A detailed introduction of the baselines can be found in the Appendix E.1.

## 4.1 Open-set node classification comparison

Since real-world scenarios are complex, where seen and unseen differs in diverse tasks, we evaluate our model in terms of open-set classification from two aspects: near open-set classification, and far open-set classification under inductive and transductive learning setting, respectively, as introduced in *test settings*.

### 4.1.1 Near open-set classification

Table 1 presents the accuracy and macro-F1 scores of the methods applied to the near open-set classification task in the inductive learning setting, where the last class of each dataset is designated as the unknown class (*i.e.*, u = 1), and the remaining classes are used for model training. It can be observed that EGonc outperforms all the baselines on the five datasets. This shows that EGonc can better differentiate between a known class and an unknown class, though they are similar to each other. Specifically, compared to the second-best method $\mathcal{G}^2Pxy$, EGonc achieves 2.41% and 2.22% improvements on average in terms of accuracy and F1 on the five datasets,

Table 2: Accuracy and macro-F1 scores of EGonc and its variants with respect to different losses and generation strategies.

| Components | | | Cora | | Citeseer | | DBLP | | Pubmed | | O_arxiv | | Average | |
|---|---|---|---|---|---|---|---|---|---|---|---|---|---|---|
| $l_1$ | $l_2$ | $l_3$ | Acc | F1 | Acc | F1 | Acc | F1 | Acc | F1 | Acc | F1 | Acc | F1 |
| ✓ | | | 84.2 | 84.7 | 75.2 | 69.0 | 76.5 | 77.7 | 70.1 | 47.3 | 61.9 | **34.1** | 73.6 | 62.6 |
| ✓ | ✓ | | 84.3 | 84.8 | 75.5 | 71.0 | 77.3 | 79.0 | 73.7 | 70.2 | 62.7 | 33.0 | 74.7 | 67.6 |
| ✓ | ✓ | ✓ | **84.5** | **84.9** | **75.8** | **71.5** | **79.1** | **80.8** | **80.2** | **75.5** | **63.0** | 33.0 | **76.5** | **69.1** |

| $X_{far}$ | $X_{rand}$ | $X_{int}$ | $X_{ext}$ | Acc | F1 | Acc | F1 | Acc | F1 | Acc | F1 | Acc | F1 | Acc | F1 |
|---|---|---|---|---|---|---|---|---|---|---|---|---|---|---|---|
| | | | | 82.7 | 83.2 | 73.5 | 69.6 | 69.5 | 71.3 | 70.4 | 67.2 | 60.1 | 30.0 | 71.2 | 64.3 |
| ✓ | | | | 83.7 | 84.0 | 75.5 | 66.9 | 72.3 | 72.7 | 71.8 | 68.5 | 62.3 | 29.3 | 73.1 | 64.3 |
| | ✓ | | | 81.3 | 82.2 | 74.6 | 63.7 | 71.2 | 71.5 | 70.0 | 66.9 | 61.9 | 32.3 | 71.8 | 63.3 |
| | | ✓ | | 84.2 | 84.7 | 75.3 | 70.8 | 75.3 | 76.9 | 73.4 | 68.7 | 62.3 | 31.4 | 74.1 | 66.5 |
| | | | ✓ | 84.1 | 84.6 | 75.4 | 70.9 | 75.5 | 74.8 | 71.4 | 66.9 | 61.5 | 29.5 | 73.6 | 65.3 |
| ✓ | ✓ | | | 84.0 | 84.4 | 75.7 | 71.2 | 72.0 | 71.7 | 73.0 | 69.1 | 61.9 | 32.0 | 73.3 | 65.7 |
| | | ✓ | ✓ | **84.5** | **84.9** | **75.8** | **71.5** | **79.1** | **80.8** | **80.2** | **75.5** | **63.0** | **33.0** | **76.5** | **69.1** |

We illustrate detailed classification accuracy in terms of known classes and unknown classes in Appendix E.4. And the results shows that in order to gain the ability of unknown class detection, there is a slight decrease in the performance of known class classification, i.e. from 79.5% to 76.3% on average, comparing EGonc to closed-set classification method GCN_soft. However, the unknown class detection accuracy is improved from 0% to 76.1% on average, which is remarkable. Besides, we also evaluate the performance of our model in terms of multiple unknown classes. The results are demonstrated in Appendix E.5 and it can be observed that EGonc consistently outperforms the baselines. We further evaluate our model in terms of near open-set classification under transductive learning setting. The results are shown in Appendix E.6. EGonc consistently performs best.

### 4.1.2 Far open-set classification

For far open-set classification, following the protocol defined in [38], the models are trained and validated by training and validation instances of the original dataset (IND data). While for testing, instances from another dataset are augmented to the original test set as open-set classes (OOD data). for example, Co_Ci means that Core as IND data and Citeseer as OOD data.

The results for are shown in Table 3. It is found that EGonc can handle far out-of-distribution classes from diverse inputs and achieve noticeable performance improvement compared to other open-set classification methods. Surprisingly, the simple thresholding approach of GCN_soft_$\tau$ achieves comparable performance with EGonc. This inspires us to combine the generative methods with discriminative methods for far open-set classification in future work.

Table 3: Accuracy and macro-F1 for far open-set classification on benchmark datasets. Numbers reported are all percentage (%).

| Methods | Co_Ci | | Ci_DB | | DB_Pub | | Average | |
|---|---|---|---|---|---|---|---|---|
| | Acc | F1 | Acc | F1 | Acc | F1 | Acc | F1 |
| GCN_soft | 43.0 | 58.9 | 38.4 | 42.5 | 41.9 | 53.7 | 41.1 | 51.7 |
| GCN_sig | 41.6 | 57.5 | 36.3 | 42.1 | 41.6 | 45.2 | 39.8 | 48.3 |
| GCN_soft_$\tau$ | 81.2 | 77.6 | 86.2 | 71.1 | 85.0 | **75.6** | 84.1 | 74.8 |
| GCN_sig_$\tau$ | 69.4 | 51.8 | 68.7 | 48.0 | 79.8 | 69.1 | 72.6 | 56.3 |
| Openmax | 56.2 | 55.1 | 69.6 | 60.3 | 69.6 | 58.7 | 65.1 | 58.0 |
| DOC | 69.4 | 57.8 | 75.5 | 62.3 | 78.0 | 70.7 | 74.3 | 63.6 |
| PROSER | 78.5 | 79.1 | 81.5 | 66.4 | 78.6 | 69.0 | 79.5 | 71.5 |
| OpenWGL | 80.6 | 76.7 | 44.6 | 11.9 | 84.6 | 70.7 | 69.9 | 53.1 |
| GNNSAFE | 79.3 | 79.9 | 80.9 | 65.9 | 80.0 | 65.0 | 80.1 | 70.3 |
| $\mathcal{G}^2 Pxy$ | 81.3 | 80.5 | 87.5 | 74.4 | 86.5 | 72.3 | 85.1 | 75.7 |
| EGonc | **81.7** | **81.0** | **88.1** | **75.2** | **87.2** | 72.8 | **85.7** | **76.3** |

### 4.2 Ablation Study

We compare variants of EGonc with respect to loss function and substitute-unknown generation strategy to demonstrate its effect. As shown in Table 2, we firstly verify the effect of each loss we adopted, i.e., $(C+1)$-class cross-entropy loss $l_1$, tailored complement entropy loss $l_2$, and energy regularization loss $l_3$. The results show that these three losses are indispensable to open-set node classification. Then we verify the effect of the substitute unknown samples we generated. Specifically,

we compare the performance of EGonc with respect to different kinds of (substitute) unknown samples: $X_{\text{far}}$ which are real far OOD samples, $X_{\text{rand}}$ which are substitute unknowns generated via $mixup$ with random parent nodes; $X_{\text{int}}$ which are generated inter-class substitute unknowns, and $X_{\text{ext}}$ which are generated external substitute unknowns. From the results, we can see that EGonc generally achieves higher accuracy with the assistance of auxiliary unknown class samples. However, well-designed unknown substitutes are most beneficial for open-set node classification.

## 4.3 Generalization Analysis

The proposed model has no specific requirement on the GNN architecture for classification. The unknown-class substitute generation strategy and energy propagation take into account the topological properties of graph data. Therefore, they efficiently achieve the generation of representative unknown class samples and perform in-depth exploration of unkown-class features across different backbones. This design contributes to the model's performance in open-set classification tasks under different backbones. Table 4 ilustrates the performance of the proposed EGonc with different GNN architectures, including GCN, GAT and GraphSage. The results confirm the effectiveness and generalization ability of EGonc for open-set node classification.

Table 4: Accuracy and macro-F1 scores of open-set classification methods with different backbone neural network. Numbers reported are all percentage (%).

| Methods | Cora | | Citeseer | | Dblp | | PubMed | | Ogbn_arxiv | |
|---|---|---|---|---|---|---|---|---|---|---|
| | Acc | F1 | Acc | F1 | Acc | F1 | Acc | F1 | Acc | F1 |
| GCN_soft_$\tau$ | 73.6 | 73.8 | 57.3 | 54.5 | 65.0 | 62.4 | 49.7 | 48.6 | 47.3 | 20.6 |
| GCN_DOC | 77.8 | 78.1 | 66.0 | 56.7 | 69.9 | 69.2 | 45.6 | 46.2 | 46.7 | 20.7 |
| GCN_Openmax | 74.6 | 75.1 | 56.2 | 54.5 | 67.2 | 67.2 | 49.1 | 48.7 | 45.5 | 16.3 |
| GCN_$\mathcal{G}^2Pxy$ | 84.3 | 84.8 | 75.5 | 71.0 | 77.3 | 79.0 | 73.7 | 70.2 | 62.7 | **33.0** |
| GCN_EGonc | **84.5** | **84.9** | **75.8** | **71.5** | **79.1** | **80.8** | **80.2** | **75.5** | **63.0** | **33.0** |
| GAT_soft_$\tau$ | 71.6 | 69.2 | 58.9 | 51.1 | 65.4 | 66.6 | 43.2 | 43.7 | 49.1 | 16.7 |
| GAT_DOC | 71.1 | 72.6 | 62.4 | 59.5 | 64.2 | 61.8 | 42.1 | 42.9 | 48.3 | 16.2 |
| GAT_Openmax | 66.3 | 63.4 | 48.6 | 48.9 | 62.5 | 56.9 | 48.6 | 47.0 | 32.2 | 8.4 |
| GAT_$\mathcal{G}^2Pxy$ | 80.4 | 81.0 | 75.2 | 70.9 | 72.9 | 73.7 | 71.7 | 47.0 | 53.7 | 22.6 |
| GAT_EGonc | **80.8** | **81.3** | **75.3** | **71.0** | **73.1** | **74.0** | **74.3** | **63.6** | **56.1** | **24.5** |
| Graphsage_soft_$\tau$ | 72.7 | 72.9 | 63.5 | 51.2 | 64.3 | 64.0 | 46.6 | 46.9 | 51.5 | 16.0 |
| Graphsage_DOC | 76.0 | 75.4 | 63.6 | 59.9 | 68.9 | 72.2 | 44.6 | 45.7 | 49.5 | 14.7 |
| Graphsage_Openmax | 71.1 | 70.6 | 47.9 | 48.7 | 62.3 | 56.9 | 44.4 | 45.1 | 43.2 | 8.0 |
| Graphsage_$\mathcal{G}^2Pxy$ | 87.2 | **87.3** | 78.6 | 76.9 | 74.4 | 74.7 | 72.8 | 64.9 | 62.8 | 36.5 |
| Graphsage_EGonc | **87.3** | **87.3** | **79.5** | **77.4** | **78.0** | **79.6** | **73.0** | **65.0** | **63.4** | **38.4** |

## 5 Limitation and Conclusion

This paper proposed a novel energy-based generative open-set node classification method, EGonc, by estimating the underlying density of the training data to decide whether a given input is close to the IND data. To obtain better OOD detection capability, we generate substitute unknowns to mimic the distribution of real open-set samples, and use energy logit to represent the indicator of OOD-ness. Under constraint of cross entropy loss, complement entropy loss, and energy regularization loss, EGonc achieves superior effectiveness for unknown class detection and known class classification, which is validated by experiments. EGonc has nice theoretical properties that guarantee an overall distinguishable margin between the detection scores for IND and OOD samples. EGonc also has good generalization since it has no specific requirement on the GNN architecture.

## Acknowledgments

This research was supported by National Natural Science Foundation of China (62206179, 92270122), Guangdong Provincial Natural Science Foundation (2022A1515010129, 2023A1515012584).

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

# A   Appendix / supplemental material

# B   Related works

## B.1   Graph Neural Networks

Graph neural networks are deep learning based models that apply neural networks to graph learning and representation [41]. They are used in a wide variety of tasks [53, 50] and have a strong theoretical foundation [39, 61]. GNNs were shown to yield state-of-the-art performance on many graph-related tasks [36, 35] and have applications in various fields such as e-commerce [32], traffic analysis [57], chemistry [33] and for knowledge bases [26].

The original GNN model brought the idea of combining the design of modern neural networks with graph learning [58], *e.g.*, Graph Convolutional Networks (GCNs) [28], Graph Attention Networks (GATs) [50], Graph Isomorphism Networks (GINs) [59], and GraphSage [16]. Then various improvements have been made to the basic GNNs for different purposes [36, 35, 69, 65]. For instance, to alleviate the over-smoothing problem, decoupling mechanisms [64] and identity mapping [4] were introduced respectively. To increase the expressive power of most GNNs by the 1-Weisfeiler-Lehman (1-WL) graph isomorphism test, identity-aware Graph Neural Networks [62] were proposed.

## B.2   Open Set Node Classification

Open set recognition and classification [1, 43] require classifiers capable of categorizing known classes and detecting objects of unknown types during testing. Two main paradigms have been investigated: generative models and discriminative models. Generative models improve prediction performance by emulating the distribution of unknown class nodes, while discriminative models use a threshold to distinguish between known and unknown classes, thereby improving prediction accuracy.

Prior methods, such as OpenMax [1] and DOC [46], mainly focus on addressing the overconfidence issue of deep neural networks for unknown classes, in image and text classification problems, respectively. Specifically, on graph structured data, OpenWGL [54] employs an uncertainty loss in the form of a graph reconstruction loss [27] on unlabeled data. PGL [37] extends a previous unsupervised domain adaption framework for the open-set scenario with graph neural networks. However, it focuses on the unsupervised domain adaption and conditional shift problem. With the development of deep learning, several novel approaches have been proposed. SRNC [72] further presents a domain adaption framework for the two most common kinds of data shifts in graph structured data, *i.e.*, open-set and closed-set data shift, by combining a graph adversarial clustering technique to shift-robust node classification. MHGL [71] extends traditional out-of-distribution (OOD) detection tasks to open-set scenarios, employing Pattern Distribution Estimator (PDE) and Multi-Hypersphere Graph Learning algorithm to model fine-grained patterns and identify previously unseen anomalous patterns. GOOD [22] proposes a weakly supervised relevance feedback approach, named Open-WRF, to mitigate sensitivity to thresholds. $\mathcal{G}^2Pxy$ [67] use the unknown proxies generated via mixup to efficiently anticipate the distribution of novel classes and achieve excellent prediction performance under the constraint of both cross entropy loss and complement entropy loss.

## B.3   Energy-based Models

Energy-based modeling as well as its application is mostly concentrated on out-of-distribution detection. Grathowohl et al. [13] found that a standard discriminative classifier of $p(y|x)$ can be interpreted as an energy-based model (EBM) [40] for the joint distribution $p(x, y)$. Junbo Zhjao, Michael Mathien et al. introduces a new model of Generative Adversarial Networks, called Energy-based Generative Adversarial Network (EBGAN) [68], which features the discriminator being viewed as an energy function, introducing a new theoretical perspective and method for Generative Adversarial Networks. Tuomas Haarnoja, Haoran Tang et al. apply EBMs to the field of reinforcement learning, and introduces a novel reinforcement learning method soft Q-learning [15] using deep energy-based polices for continuous states and actions, focusing on maximum entropy policies and improved task exploration. Yilun Du, Joshua Meier et al. innovatively applies an energy-based model [7] to score protein conformations at the atomic level, introducing machine learning methods into the

field of protein design. Ruiqi Gao, Yang Song et al. introduces a method called Diffusion Recovery Likelihood [11] to effectively learn and sample from Energy-Based Models (EBMs) by learning from the diffused versions of the data, overcoming the challenges of training and sampling EBMs on high-dimensional datasets. MareLafun, Elias Ramzi et al. introduced the HEAT [29], which leverages the versatility of the EBM framework to provide a strong OOD detection method. Qitian Wu, Yiting Chen et al. applied EBM to the task of OOD detection in graph domains and proposed GNNSAFE [56]. GNNSAFE is a method for detecting anomalies in graph data, which can improve the robustness and generalization of GNN. Impressively, the use of unknown classes samples information to improve EBM performance is discussed in GNNSAFE.

## C   Proofs for Proposition

### C.1   Proof for Proposition 1

The gradient of $l_1$ is given by

$$
\begin{aligned}
\frac{\partial l_1}{\partial \theta} &= \sum_{i \in \mathcal{D}_{\mathrm{tr}}} \left( -\frac{\partial h_i^k(\theta; x_i, A)_{[y_i]}}{\partial \theta} + \sum_{c=1}^{C+1} \frac{\partial h_i^k(\theta; x_i, A)_{[c]}}{\partial \theta} \frac{e^{h_i^k(\theta; x_i, A)_{[c]}}}{\sum_{c'=1}^{C+1} e^{h_i^k(\theta; x_i, A)_{[c']}}} \right) \\
&= \sum_{i \in \mathcal{D}_{\mathrm{tr}}} \left( \frac{\partial E(x_i, A, y_i)}{\partial \theta} - \sum_{c=1}^{C+1} \frac{\partial E(x_i, A, c)}{\partial \theta} \frac{e^{h_i^k(\theta; x_i, A)_{[c]}}}{\sum_{c'=1}^{C+1} e^{h_i^k(\theta; x_i, A)_{[c']}}} \right) \\
&= (1 - p(y = y_i \mid \mathbf{x}_i, A)) \frac{\partial E(\mathbf{x}_i, A, y_i; h_\theta)}{\partial \theta} - \sum_{c \neq y_i} p(y = c \mid \mathbf{x}_i, A) \frac{\partial E(\mathbf{x}_i, A, c; h_\theta)}{\partial \theta}.
\end{aligned}
\tag{15}
$$

The gradient of $l_2$ is given by

$$
\begin{aligned}
\frac{\partial l_2}{\partial \theta} &= \sum_{i \in \mathcal{D}_{\mathrm{tr}}} \left( -\frac{\partial h_i^k(\theta; x_i, A)_{[C+1]}}{\partial \theta} + \sum_{j \neq y_i}^{C+1} \frac{\partial h_i^k(\theta; x_i, A)_{[j]}}{\partial \theta} \frac{e^{h_i^k(\theta; x_i, A)_{[j]}}}{\sum_{c' \neq y_i}^{C+1} e^{h_i^k(\theta; x_i, A)_{[c']}}} \right) \\
&= \sum_{i \in \mathcal{D}_{\mathrm{tr}}} \left( \frac{\partial E(x_i, A, C+1)}{\partial \theta} - \sum_{j \neq y_i}^{C+1} \frac{\partial E(x_i, A, j)}{\partial \theta} \frac{e^{h_i^k(\theta; x_i, A)_{[j]}}}{\sum_{c' \neq y_i}^{C+1} e^{h_i^k(\theta; x_i, A)_{[c']}}} \right) \\
&= (1 - p'(y = C+1 \mid \mathbf{x}_i, A)) \frac{\partial E(\mathbf{x}_i, A, C+1; h_\theta)}{\partial \theta} - \sum_{j \neq y_i \cap j \neq C+1} p'(y = j \mid \mathbf{x}_i, A) \frac{\partial E(\mathbf{x}_i, A, j; h_\theta)}{\partial \theta}.
\end{aligned}
\tag{16}
$$

where $p'(y = j \mid \mathbf{x}_i, A) = \frac{e^{h_i^k(\theta; x_i, A)_{[j]}}}{\sum_{c' \neq y_i}^{C+1} e^{h_i^k(\theta; x_i, A)_{[c']}}}$ is an approximate probability distribution established on the complementary classes

In the Eq. (15) and Eq. (16), We computed the gradient with respect to the parameters. The information presented in the preceding equation indicates that the training process for $\mathcal{D}_{\mathrm{tr}}$, which aims to minimize the first-order gradient of $l_1 + \lambda_2 l_2$, leads to a reduction in the energy score. In a broader context, the energy output generated by the trained model tends to decrease for any instance originating from the distribution $\mathcal{D}_{\mathrm{tr}}$. Besides, for the unknown substitutes $X_{\mathrm{sub}}$ we constructed, we will adjust their energy scores by the $l_3$ to keep the energy model in an optimal state for fulfilling its function.

## C.2 Proof for Proposition 2

We can demonstrate the proposition 2 when considering $\frac{\sum_j \hat{A}_{ij}}{E}^{(k-1)}_j \sum_j \hat{A}_{ij} < E_i^{(k-1)}$. We need to rewrite the Eq. (10) from the perspective of a single vector:

$$
\begin{aligned}
E_i^{(k)} &= \alpha E_i^{(k-1)} + (1-\alpha)\frac{\sum_j \hat{A}_{ij}}{E}^{(k-1)}_j \sum_j \hat{A}_{ij} \\
&< \alpha E_i^{(k-1)} + (1-\alpha)E_i^{(k-1)} \\
&= E_i^{(k-1)}.
\end{aligned}
\tag{17}
$$

A similar prove can be applied to establish the contrary outcome in the case $\frac{\sum_j a_{ij} E_j^{(k-1)}}{\sum_j \hat{A}_{ij}} > E_i^{(k-1)}$

$$
\begin{aligned}
E_i^{(k)} &= \alpha E_i^{(k-1)} + (1-\alpha)\frac{\sum_j \hat{A}_{ij} E_j^{(k-1)}}{\sum_j \hat{A}_{ij}} \\
&> \alpha E_i^{(k-1)} + (1-\alpha)E_i^{(k-1)} \\
&= E_i^{(k-1)}.
\end{aligned}
\tag{18}
$$

## C.3 Proof for Proposition 3

At first, we define $E(x; h_{\theta^*})$ as the energy scores yielded by the model $h_{\theta^*}$ minimizing the energy regularization loss $l_3$. Then, $h_{\theta_1^+}$ and $h_{\theta_2^+}$ denotes the model yielding the optimal predictive softmax distribution that minimizing the cross entropy loss $l_1$ and the model yielding the optimal predictive softmax distribution that minimizing the tailored complement entropy loss $l_2$. *i.e.*,

$$
\frac{e^{h_{\theta_1^+}(x, \mathcal{G}_x)_{[y]}}}{\sum_{c=1}^{C+1} e^{h_{\theta_1^+} x_{[c]}}} = argmin_{p(y|x,\mathcal{G}_x)}\mathbb{E}_{(x,\mathcal{G}_x,y)\in D_{tr}}[-log\, p(y|x,\mathcal{G})]
\tag{19}
$$

$$
\frac{e^{h_{\theta_2^+}(x, \mathcal{G}_x)_{[C+1]}}}{\sum_{c\neq y}^{C+1} e^{h_{\theta_2^+} x_{[c]}}} = argmin_{p(C+1|x/y,\mathcal{G}_x)}\mathbb{E}_{(x,\mathcal{G}_x,y)\in D_{tr}}[-log\, p(C+1|x/y,\mathcal{G})]
\tag{20}
$$

At the same time, considering that $E(x, \mathcal{G}_x; h_\theta) = -log\sum_{c=1}^{C+1} e^{h_\theta(x,\mathcal{G}_x)}$. Therefore we start from this equation and we have

$$
\begin{aligned}
E(x, \mathcal{G}_x, h_{\theta^*}) &= E(x, \mathcal{G}_x; h_{\theta^*}) - E(x, \mathcal{G}_x; h_{\theta_1^+}) - log\sum_{c=1}^{C+1} e^{h_{\theta_1^+}(x,\mathcal{G}_x)} \\
&= -log(e^{-E(x,\mathcal{G}_x;h_{\theta^*}) + E(x,\mathcal{G}_x;h_{\theta_1^+})} \cdot \sum_{c=1}^{C+1} e^{h_{\theta_1^+}(x,\mathcal{G}_x)}) \\
&= -log\sum_{c=1}^{C+1} e^{h_{\theta_1^+}(x,\mathcal{G}_x)-E(x,\mathcal{G}_x;h_{\theta^*})+E(x,\mathcal{G}_x;h_{\theta_1^+})}.
\end{aligned}
\tag{21}
$$

The above equation implies an equivalence relationship between $E(x, \mathcal{G}_x, h_{\theta^*})$ and $E(x, \mathcal{G}_x; h_{\theta^*}) - E(x, \mathcal{G}_x; h_{\theta_1^+}) - log\sum_{c=1}^{C+1} e^{h_{\theta_1^+}(x,\mathcal{G}_x)}$. Meanwhile, continuing to consider that $E(x, \mathcal{G}_x; h_{\theta^*}) = -log\sum_{c=1}^{C+1} e^{h_{\theta^*}(x,\mathcal{G}_x)}$, we can further demonstrate that the predictive softmax probability distribution obtained from $E(x, \mathcal{G}_x, h_{\theta^*})$ is

$$p(y|x,\mathcal{G}_x) = \frac{e^{h_{\theta_1^+}(x,\mathcal{G}_x)_{[y]}-E(x,\mathcal{G}_x;h_{\theta^*})+E(x,\mathcal{G}_x;h_{\theta_1^+})}}{\sum_{c=1}^{C+1} e^{h_{\theta_1^+}(x,\mathcal{G}_x)_{[c]}-E(x,\mathcal{G}_x;h_{\theta^*})+E(x,\mathcal{G}_x;h_{\theta_1^+})}}$$

$$= \frac{e^{h_{\theta_1^+}(x,\mathcal{G}_x)_{[y]}} \cdot e^{-E(x,\mathcal{G}_x;h_{\theta^*})+E(x,\mathcal{G}_x;h_{\theta_1^+})}}{\left(\sum_{c=1}^{C+1} e^{h_{\theta_1^+}(x,\mathcal{G}_x)_{[c]}}\right) \cdot e^{-E(x,\mathcal{G}_x;h_{\theta^*})+E(x,\mathcal{G}_x;h_{\theta_1^+})}} \qquad (22)$$

$$= \frac{e^{h_{\theta_1^+}(x,\mathcal{G}_x)_{[y]}}}{\sum_{c=1}^{C+1} e^{h_{\theta_1^+}(x,\mathcal{G}_x)_{[c]}}}.$$

The above predictive softmax probability distribution exactly minimizes $l_1$ as defined by Eq (19). We thus have proven that the optimal energy that minimizes $l_3$ intrinsically induce the predictive softmax distribution that minimizes $l_1$ for $D_{\text{tr}}$.

Similarly, when we consider the relationship between the $l_2$ and $l_3$, we can start again from $E(x,\mathcal{G}_x;h_\theta) = -log\sum_{c=1}^{C+1} e^{h_\theta(x,\mathcal{G}_x)}$ and get

$$E(x,\mathcal{G}_x,h_{\theta^*}) = -log \sum_{c=1}^{C+1} e^{h_{\theta_2^+}(x,\mathcal{G}_x)-E(x,\mathcal{G}_x;h_{\theta^*})+E(x,\mathcal{G}_x;h_{\theta_2^+})}. \qquad (23)$$

The above equation implies an equivalence relationship between $E(x,\mathcal{G}_x,h_{\theta^*})$ and $E(x,\mathcal{G}_x;h_{\theta^*}) - E(x,\mathcal{G}_x;h_{\theta_2^+}) - log\sum_{c=1}^{C+1} e^{h_{\theta_2^+}(x,\mathcal{G}_x)}$. We thus can further demonstrate that the predictive softmax probability distribution with respect to class C + 1 obtained from $E(x,\mathcal{G}_x,h_{\theta^*})$ is

$$p(C+1|x,\mathcal{G}_x) = \frac{e^{h_{\theta_2^+}(x,\mathcal{G}_x)_{[C+1]}-E(x,\mathcal{G}_x;h_{\theta^*})+E(x,\mathcal{G}_x;h_{\theta_2^+})}}{\sum_{c=1}^{C+1} e^{h_{\theta_2^+}(x,\mathcal{G}_x)_{[c]}-E(x,\mathcal{G}_x;h_{\theta^*})+E(x,\mathcal{G}_x;h_{\theta_2^+})}}$$

$$= \frac{e^{h_{\theta_2^+}(x,\mathcal{G}_x)_{[C+1]}} \cdot e^{-E(x,\mathcal{G}_x;h_{\theta^*})+E(x,\mathcal{G}_x;h_{\theta_2^+})}}{\left(\sum_{c=1}^{C+1} e^{h_{\theta_2^+}(x,\mathcal{G}_x)_{[c]}}\right) \cdot e^{-E(x,\mathcal{G}_x;h_{\theta^*})+E(x,\mathcal{G}_x;h_{\theta_2^+})}}$$

$$= \frac{e^{h_{\theta_2^+}(x,\mathcal{G}_x)_{[C+1]}}}{\sum_{c\neq y}^{C+1} e^{h_{\theta_2^+}(x,\mathcal{G}_x)_{[c]}}} \cdot \frac{\sum_{c\neq y}^{C+1} e^{h_{\theta_2^+}(x,\mathcal{G}_x)_{[c]}}}{\sum_{c=1}^{C+1} e^{h_{\theta_2^+}(x,\mathcal{G}_x)_{[c]}}} \qquad (24)$$

$$= \left(1 - \frac{e^{h_{\theta_2^+}(x,\mathcal{G}_x)_{[y]}}}{\sum_{c=1}^{C+1} e^{h_{\theta_2^+}(x,\mathcal{G}_x)_{[c]}}}\right) \cdot \frac{e^{h_{\theta_2^+}(x,\mathcal{G}_x)_{[C+1]}}}{\sum_{c\neq y}^{C+1} e^{h_{\theta_2^+}(x,\mathcal{G}_x)_{[c]}}}$$

Considering that the energy scores are not computed for a specific class of labels, but rather are computed considering all classes, and the loss $l_3$ tends to reduce the energy scores for $x \in D_{\text{tr}}$. Meanwhile, combine this with the fact that the loss $l_2$ tends to drive a higher probability of output for class $C$, it is reasonable to assume that the probability of this probability distribution is not overly large for class $y$. Then, The former term can be regarded as a coefficient modifying the latter term. We thus have proven that the optimal energy that minimizes $l_3$ intrinsically induce the predictive softmax distribution that minimizes $l_2$ to a certain extent for $D_{\text{tr}}$.

## D  Algorithm

The algorithm of EGonc is illustrated in Algorithm 1. The given graph data are inputted into a GNN model (Line 4), followed by the generation of inter-class and external unknown substitutes. Then the generated substitutes and the original training data are input into the rest of neural network and learn an open-set classifier (Lines 6-10). As new graphs are fed in for training (Line 6), the energy scores for the known classes samples and unknown class samples are calculated (Line 7). Under the constraint of the total loss, the model is trained and optimized.

---

**Algorithm 1** EGonc: open-set node classification

---

**Require:** : $G = (V, E, X)$: a graph with links and features;

$\mathcal{D}_{\text{tr}} = \{G, Y\}$: train set with labeled nodes;

$X_{\text{te}} = S \cup U$: test set where $S$ are the known classes appeared in training and $U$ are the unknown classes;

**Ensure:** $f(X_{\text{te}} \to \mathcal{Y}), \mathcal{Y} \in \{1, \dots, C, unknown\}$.

1: Obtain the inter-class node pairs $\{(x_i, y_i), (x_j, y_j)\} \in \mathcal{D}_{\text{tr}}$ $s.t.$ $y_i \neq y_j \& a_{ij} = 1$
2: Obtain the peripheral nodes that are leaf nodes and low confident nodes.
3: **while** not convergence **do**
4:     For the first $m = 1, \dots k$ layer:
       $h_i^m = f^m(\theta_1; h_i^{m-1}, h_j^{m-1}, j \in \mathcal{N}_i), \forall x_i \in \mathcal{D}_{\text{tr}}$
5:     At the $k$-th layer:
       Create unknown substitutes $X_{\text{sub}}$ using Eq. (8) & (9)
       Augment the substitutes to known class samples:
       $\overline{\mathcal{D}}_{\text{tr}} = \mathcal{D}_{\text{tr}} \cup (X_{\text{sub}}, Y_{\text{C+1}})$
6:     For the $m = k + 1, \dots, k_1 - 1$ layers:
       $h_i^m = f^m(\theta_1; h_i^{m-1}, h_j^{m-1}, j \in \mathcal{N}_i), \forall x_i \in \overline{\mathcal{D}}_{\text{tr}}$
7:     For the $m = k_1, \dots, \mathcal{K}$ layers:
       $h_i^m = f^m(\theta_1; h_i^{m-1}, h_j^{m-1}, j \in \mathcal{N}_i), \forall x_i \in \overline{\mathcal{D}}_{\text{tr}}$
       $E_i^m = f^m(E_i^{m-1}, E_j^{m-1}, j \in \mathcal{N}_i), \forall x_i \in \overline{\mathcal{D}}_{\text{tr}}$
8:     For open-set classifier layer:
       Obtain cross entropy loss as Eq. (11)
       Obtain complement entropy loss as Eq. (12)
       Obtain energy regularization loss as Eq. (13)
9:     Back-propagate loss gradient using Eq. (14) and update weights
10:    **if** early stopping condition is satisfied **then**
11:        Terminate
12:    **end if**
13: **end while**

---

## D.1 Complexity Analysis

Detailed training process of EGonc is illustrated in Algorithm 1. In terms of complexity, compared with normal closed-set node classification, our extra computation comes from three parts: 1) unknown substitute generation, which has linear complexity with the quantity of edges and the quantity of generated substitutes, $i.e.$, $\mathcal{O}(|E| + |X_{\text{sub}}|)$; 2) energy scores calculation, which has linear complexity with the quantity of the known class samples after augmentation, $i.e.$, $\mathcal{O}(\overline{\mathcal{D}}_{\text{tr}})$ 3) substitute classifier training, the complexity of this part depends on the adopted backbone GNN and the quantity of generated substitutes. Assume $\mathcal{O}(|f_\theta|)$ be the time GNN $f_{\theta_1}$ and classifier $f_{\theta_2}$ takes to compute a single node embedding and make a prediction, the complexity of substitute classifier training is $\mathcal{O}(|X_{\text{sub}}| \cdot |f_\theta|)$. Overall, the extra complexity is linear with the number of edges in the original graph and the number of generated substitutes. Experimental results on parameter analysis of generated substitutes quantity (see Fig. 1) show that a relatively small number of substitutes can achieve quite good performance. Thus, the extra complexity is reasonable for open-set node classification.

# E Supplementary experiments

## E.1 Baseline Details

- GCN_soft, GCN_sig : GCN [28] is adopted for graph learning. We constructed four baselines using GCN, where GCN_soft and GCN_sig use softmax and sigmoid layers, respectively, as the final layer for training and classification. Here we adopt GCN with sigmod layer as baselines since softmax is observed to be overconfident to unknown class samples.

- GCN_soft_$\tau$, GCN_sig_$\tau$: Additionally, GCN_soft_$\tau$ and GCN_sig_$\tau$ perform classification based on these two model using fixed thresholds chosen in $\{0.1, 0.2, \dots, 0.9\}$.

- Openmax: Openmax [1] is an open-set recognition model based on "activation vectors", $i.e.$, penultimate layer of the network.

By modeling the distance from "mean activation vector" using the extreme value distribution, softmax scores are calibrated for each class and updated to Openmax scores, which are then used for open-set classification. GCN is further combined with Openmax for open-set node classification on graph data.

- DOC: DOC [46] is an open-world classification method for text classification. It uses multiple 1-vs-rest of sigmoids as the final output layer and defines an automatic threshold setting mechanism. GCN is combined with DOC to allow comparison with the proposed model. GCN is used to obtain the node representations and DOC is used to do open-set classification and rejection.
- PROSER: PROSER [70] is a novel and highly effective open-set recognition method used in the field of image processing. We paired it with GCN for graph-based comparison.
- OpenWGL: OpenWGL [54] employs an uncertainty loss in the form of graph reconstruction loss on unlabeled data and using an adaptive threshold to detect the unknown class samples.
- GNNSAFE : GNNSAFE [56] proposes an effective OOD discriminator based on an energy function derived from graph neural networks (GNNs) trained with standard classification loss. We apply it from the OOD detection domain to the open-set classification domain.
- $\mathcal{G}^2Pxy$ : $\mathcal{G}^2Pxy$ [67] uses the generated proxies generated via mixup, with the help of cross entropy loss and complement entropy loss, shows excellent performance in open-set classification

### E.2 Datasets Stastics

Table 5: Statistics of the experimental datasets.

| Dataset | Nodes | Edges | Features | Labels |
|---|---|---|---|---|
| Cora | 2708 | 5429 | 1433 | 7 |
| Citeseer | 3312 | 4732 | 3703 | 6 |
| DBLP | 17716 | 105734 | 1639 | 4 |
| PubMed | 19717 | 44325 | 500 | 3 |
| Ogbn_arxiv | 169343 | 1166243 | 128 | 40 |

### E.3 Implementation Details

The GCN is configured with two hidden GCN layers in the dimension of 512 and 128, followed by an additional multilayer perceptron layer of size 64. EGonc is implemented with PyTorch and the networks are optimized using stochastic gradient descent with a learning rate of $1e^{-3}$. The balance parameters $\lambda_1$, $\lambda_2$ and $\lambda_3$ are chosen by a grid search in the interval from $10^{-2}$ to $10^2$ with a step size of $10^1$.

The baseline methods are evaluated according to the instructions reported in the original papers with the same parameter configuration unless otherwise specified, and the best results are selected. For each experiment, the baselines and the proposed method were applied using the same training, validation, and testing datasets. The hyperparameters were tuned to get the best performance on the validation set. All the experiments were conducted on a workstation equipped with an Intel(R) Xeon(R) Gold 6226R CPU and an Nvidia A100.

### E.4 Detailed performance on *IND* and *OOD* classes for near open-set classification

We show the detailed classification accuracy in terms of known classes (IND classes) and unknown classes (OOD classes) respectively, in Table 6. The experiment was conducted under the same setting of Table 1. It shows that in order to gain the ability of unknown class detection, there is a slight decrease in the performance of known class classification, i.e. from 79.5% to 76.3% on average, comparing EGonc to closed-set classification method GCN_soft. However, the unknown class detection accuracy is improved from 0% to 76.1% on average, which is remarkable. Compared to other open-set classification methods, such as the globally second best method, $\mathcal{G}^2Pxy$, the average performance of unknown-class node detection is improved from 70.7% to 76.1% while the performance of the known-class classification experienced a slight decrease from 78.6% to 76.3% on average. When compared with the globally third best method, PROSER, the average performance of unknown-class node detection is improved from 63.6% to 76.1% while the performance of the known-class classification still gain a certain increase from 73.0% to 76.3% on average.

Table 6: Detailed classification accuracy in terms of known classes ( in-distribution, *ind*) and unknown classes (out-of-distribution, *ood*) for near open-set classification on five datasets with one unknown class ($u = 1$) under inductive learning. Numbers reported are all percentage (%).

| Methods | Cora | | Citeseer | | DBLP | | Pubmed | | Ogbn_arxiv | | Average | | |
|---|---|---|---|---|---|---|---|---|---|---|---|---|---|
| | ind | ood | ind | ood | ind | ood | ind | ood | ind | ood | ind | ood | overall |
| GCN_soft | **89.7** | 00.0 | 71.8 | 00.0 | **92.9** | 00.0 | **93.2** | 00.0 | 50.1 | 00.0 | **79.5** | 00.0 | 51.5 |
| GCN_sig | 87.8 | 00.0 | 73.0 | 00.0 | 92.5 | 00.0 | 93.0 | 00.0 | 49.1 | 00.0 | 79.1 | 00.0 | 51.1 |
| GCN_soft_$\tau$ | 87.2 | 23.3 | 61.4 | 50.5 | 85.6 | 19.2 | 70.4 | 40.3 | 47.4 | 35.3 | 70.4 | 33.7 | 58.6 |
| GCN_sig_$\tau$ | 85.6 | 57.9 | 67.1 | 53.8 | 79.9 | 45.8 | 78.1 | 30.2 | 46.2 | 10.0 | 71.4 | 39.5 | 60.4 |
| Openmax | 86.6 | 30.0 | 60.4 | 49.4 | 85.4 | 27.6 | 73.4 | 38.1 | 45.7 | 19.3 | 70.3 | 32.9 | 58.5 |
| DOC | 84.0 | 54.9 | 67.7 | 63.2 | 79.8 | 48.1 | 78.8 | 30.7 | 46.8 | 32.3 | 71.4 | 45.8 | 61.2 |
| PROSER | 83.2 | 82.7 | 72.5 | 76.0 | 78.2 | 58.0 | 78.0 | 70.2 | 53.1 | 31.2 | 73.0 | 63.6 | 70.5 |
| OpenWGL | 83.4 | 58.6 | 66.4 | 59.9 | 76.6 | 60.0 | 87.9 | 55.2 | 45.3 | 58.7 | 71.9 | 58.5 | 64.9 |
| GNNSAFE | 86.6 | 53.3 | 74.0 | 62.4 | 85.6 | 43.8 | 92.2 | 60.1 | 51.3 | 33.1 | 77.9 | 50.5 | 68.6 |
| $\mathcal{G}^2Pxy$ | 83.8 | **86.5** | 73.6 | **78.7** | 85.1 | 60.3 | 87.6 | 67.5 | 62.8 | 60.3 | 78.6 | 70.7 | 74.7 |
| EGonc | 84.3 | 84.9 | **74.5** | 78.1 | 79.4 | **80.7** | 80.3 | **80.1** | **63.0** | **69.1** | 76.3 | **76.1** | **76.5** |

Table 7: Near open-set classification accuracy on three datasets with multiple unknown classes under the inductive learning setting. Numbers reported are all percentage (%).

| Methods | Cora | | | | Citeseer | | | | Ogbn_arxiv | | | | | | | |
|---|---|---|---|---|---|---|---|---|---|---|---|---|---|---|---|---|
| | (u=2) | | (u=3) | | (u=2) | | (u=3) | | (u=5) | | (u=10) | | (u=15) | | (u=20) | |
| | Acc | F1 | Acc | F1 | Acc | F1 | Acc | F1 | Acc | F1 | Acc | F1 | Acc | F1 | Acc | F1 |
| GCN_soft | 48.6 | 50.3 | 34.2 | 42.6 | 28.0 | 31.6 | 14.8 | 18.1 | 48.7 | 15.0 | 47.7 | 17.6 | 42.4 | 17.4 | 29.5 | 13.7 |
| GCN_sig | 49.5 | 50.7 | 34.8 | 41.7 | 27.3 | 32.1 | 14.8 | 17.7 | 49.4 | 12.2 | 45.4 | 9.4 | 42.3 | 9.7 | 26.6 | 9.3 |
| GCN_soft_$\tau$ | 69.9 | 68.3 | 64.3 | 65.6 | 61.7 | 51.3 | 61.1 | 39.2 | 44.3 | 15.0 | 48.2 | 17.8 | 49.0 | 17.3 | 56.2 | 14.1 |
| GCN_sig_$\tau$ | 71.6 | 68.9 | 65.2 | 53.2 | 63.7 | 48.9 | 57.6 | 44.9 | 46.4 | 11.9 | 48.2 | 9.9 | 47.4 | 7.0 | 61.0 | 9.0 |
| Openmax | 61.2 | 52.7 | 56.2 | 58.4 | 49.7 | 45.7 | 43.0 | 32.9 | 42.6 | 14.3 | 47.6 | 17.3 | 48.3 | 17.6 | 54.4 | 10.0 |
| DOC | 71.8 | 69.9 | 62.6 | 56.9 | 71.8 | 53.4 | 68.8 | 47.4 | 45.3 | 15.2 | 46.6 | 16.7 | 46.8 | 7.0 | 60.2 | 12.0 |
| PROSER | 80.6 | 80.9 | 75.8 | 72.4 | 73.7 | 63.4 | 68.0 | 53.7 | 60.4 | 27.1 | 53.2 | 22.6 | 54.6 | 30.1 | 53.3 | 19.2 |
| OpenWGL | 73.1 | 71.6 | 69.3 | 66.9 | 73.5 | 41.0 | 62.5 | 47.7 | 46.8 | 20.1 | 53.4 | 21.9 | 56.7 | 22.6 | 65.2 | 9.1 |
| GNNSAFE | 78.3 | 80.5 | 80.8 | 66.9 | **79.4** | 60.6 | 76.3 | **57.7** | 55.8 | 25.7 | 51.9 | 24.7 | 53.9 | 18.9 | 59.4 | 19.7 |
| $\mathcal{G}^2Pxy$ | 82.2 | 83.4 | 80.2 | 68.9 | 77.3 | 65.3 | 79.4 | 55.5 | 68.9 | 46.1 | 55.1 | 26.5 | 63.3 | **30.4** | 65.3 | 23.9 |
| EGonc | **83.3** | **83.9** | **81.3** | **70.9** | 77.6 | **65.6** | **80.0** | 55.7 | **69.5** | **46.7** | **55.2** | **26.8** | **63.8** | 30.3 | **65.5** | **27.3** |

## E.5 Near open-set classification on multiple unknown classes

Further, we explore the performance in terms of multiple unknown classes. In Table 7, we provide the results of inductive learning on near open-set classification with multiple unknown classes, *i.e.*, u = 2,3 for the Cora and Citeseer datasets and u=5,10,15,20 for ogbn_arxiv dataset, respectively. It can be observed that EGonc consistently performs best.

## E.6 Near open-set classification under transductive learning setting

Table 8 presents the accuracy and F1 scores of the methods applied to the near open-set classification task with one unknown class (the last class), under the transductive learning setting, *i.e.*, the features of unknown class nodes are used to facilitate the model training or validation. It is observed that EGonc still consistently outperforms the baselines. Specifically, compared with the second-best method *i.e.*, $\mathcal{G}^2Pxy$, EGonc achieves 3.11% and 2.89% improvements on average in terms of accuracy and F1, and achieves 9.94% and 13.6% improvements over the third-best method (PROSER) on average in terms of accuracy and F1, respectively. Furthermore, by comparing Table 1 and Table 8, it is observed that the involving of real unknown class information during training or validation benefits the model, which is reasonable.

## E.7 Parameter learning on quantity of substitutes.

The substitute unknown node generation module of EGonc is crucial and the quantity of generated substitutes need to be predefined. We assess the performance of EGonc with different numbers of

Table 8: Near open-set classification on five citation network datasets with one unknown class (u=1) under in the *transductive learning setting*. Numbers reported are all percentage (%).

| Methods | Cora | | Citeseer | | DBLP | | Pubmed | | Ogbn_arxiv | | Average | |
|---|---|---|---|---|---|---|---|---|---|---|---|---|
| | Acc | F1 | Acc | F1 | Acc | F1 | Acc | F1 | Acc | F1 | Acc | F1 |
| GCN_soft | 70.8 | 68.2 | 44.7 | 38.9 | 62.9 | 57.0 | 29.2 | 29.7 | 50.2 | 18.4 | 51.6 | 42.4 |
| GCN_sig | 68.8 | 64.5 | 44.6 | 40.1 | 63.4 | 59.2 | 29.0 | 29.5 | 46.8 | 8.4 | 50.5 | 40.3 |
| GCN_soft_$\tau$ | 78.1 | 78.9 | 67.3 | 57.0 | 67.3 | 67.7 | 68.9 | 27.2 | 49.6 | 19.0 | 66.2 | 50.0 |
| GCN_sig_$\tau$ | 78.3 | 78.5 | 65.4 | 55.3 | 71.4 | 71.5 | 69.0 | 27.2 | 45.9 | 7.7 | 66.0 | 48.0 |
| Openmax | 77.2 | 76.9 | 57.5 | 56.7 | 69.0 | 70.6 | 55.0 | 52.1 | 49.2 | 18.9 | 61.6 | 55.0 |
| DOC | 77.3 | 77.9 | 65.1 | 55.3 | 71.7 | 72.0 | 68.4 | 34.2 | 49.9 | 19.4 | 66.5 | 51.8 |
| PROSER | 84.7 | 83.6 | 74.3 | 66.6 | 75.3 | 71.6 | 72.8 | 60.8 | 55.0 | 30.7 | 72.4 | 62.7 |
| OpenWGL | 83.3 | 83.5 | 70.0 | 65.4 | 74.3 | 74.2 | 71.2 | 68.0 | 46.0 | 20.0 | 69.0 | 62.2 |
| GNNSAFE | 80.7 | 81.9 | 73.1 | 62.2 | 74.2 | 75.8 | 73.5 | 69.9 | 52.8 | 24.1 | 70.8 | 62.8 |
| $\mathcal{G}^2Pxy$ | 90.7 | 89.7 | 76.3 | 71.8 | 77.5 | 79.5 | 78.0 | 73.4 | 63.7 | 31.4 | 77.2 | 69.2 |
| EGonc | **91.2** | **90.4** | **77.2** | **72.9** | **79.4** | **80.7** | **86.5** | **80.5** | **63.8** | **31.6** | **79.6** | **71.2** |

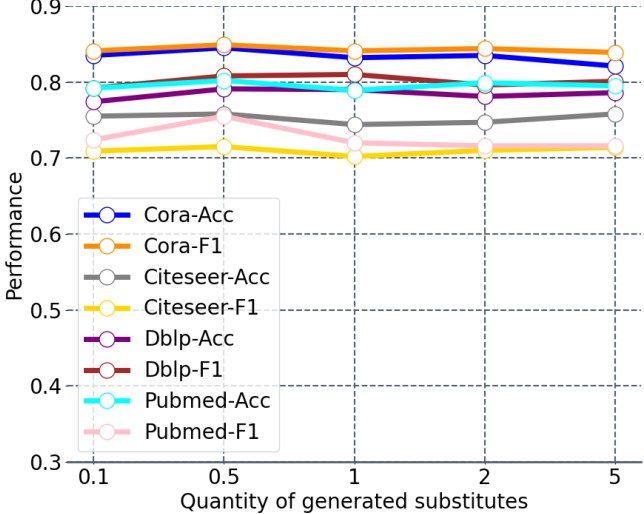

Figure 1: Performance of EGonc under different quantity of substitutes.

generated substitutes, *i.e.*, $\{0.1, 0.5, 1, 2, 5\}$ times the average number of nodes in each known class, to see the influence of the quantity of substitutes. Results are shown in Fig. 1. It is observed that both accuracy and macro-F1 are stable while the quantity of generated substitutes varies a lot. Thus, a relatively small number of substitutes can be used for better efficiency.

### E.8 Parameter learning on weights of losses.

In ablation study, we verify that the cross-entropy loss (Eq. (11)), tailored complement entropy loss (Eq. (12)), and energy regularization loss (Eq. (13)) are all indispensable. Here we further assess the performance of EGonc with various weights of these three losses, *i.e.*, $\lambda_1, \lambda_2$, and $\lambda_3$, with the range of $\{0.01, 0.1, 1, 10, 100\}$. The results are shown in Fig. 2. We can see that it is better to choose small values for the weights generally. And the wights for tailored complement entropy loss ($\lambda_2$) are relatively large compared to the weights of other two losses ($\lambda_1, \lambda_3$). For example, the best value for $\lambda_2$ is 1 for Cora, Citeseer, Dblp and 10 for Pubmed while the best values for $\lambda_1$ and $\lambda_3$ are either 0.01 or 0.1. It illustrates the importance of complement entropy loss in open-set node classification task.

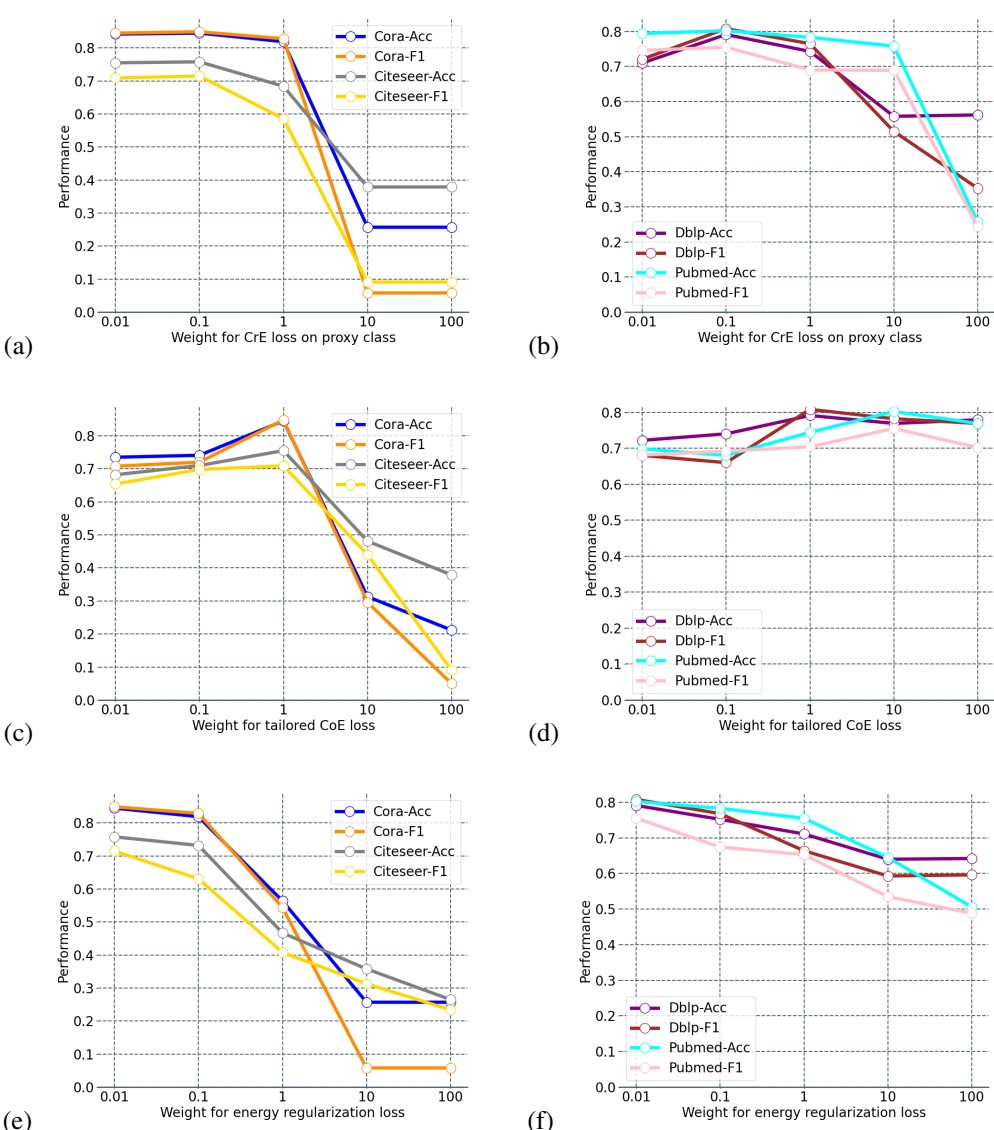

Figure 2: Performance of EGonc with different value of weights for losses we adopted.

