# OpenReview forum: "EGonc : Energy-based Open-Set Node Classification with substitute Unknowns"
_NeurIPS.cc/2024/Conference — NeurIPS 2024 poster_

### Official Review · Reviewer_exuD · 2024-06-30

**Soundness:** 1
**Presentation:** 1
**Contribution:** 2
**Rating:** 5
**Confidence:** 4

**Summary:**

This paper introduces a new method EGonc for open-set node classification in graphs. EGonc performs open-set classification by incorporating energy-based models into the graph learning diagram. The training of the model only requires in-distribution data, while out-of-distribution training data are generated synthetically. In the experiments, the authors compare EGonc against a wide range of baseline methods on several benchmarks and provide ablation studies.

**Strengths:**

1. The paper focuses on open-set node classification in graph learning, which is an important problem in real-world applications.
2. Incorporating energy-based models into graph learning is a new method that has not been explored.
3. The proposed method shows good performance, beating a wide range of baselines as is shown in the experiments.

**Weaknesses:**

1. An important claim made in the paper is not supported. The paper claims that the proposed method "has nice theoretical properties that guarantee an overall distinguishable margin between the detection scores for IND and OOD samples", but there is no theoretical analysis supporting this claim.

2. The experiments are not sufficient. First, using one class as the out-of-distribution (OOD) class is very different from the real-world scenario where OOD examples can be quite diverse. Second, the paper does not compare against several important OOD detection baselines[1][2]. The proposed method has a negative impact on the in-distribution accuracy, while the methods in [1][2] will not. Third, the error bars are not reported. Fourth, an ablation study for energy propagation is missing, which seems to be an important component of the method.

3. The presentation needs major improvement. First, there are a lot of format issues (e.g., missing reference in line 127, texts in the equations are not corrected formatted). Second, the intuitions behind $l_2$ and $l_3$ are not well-explained. It is not clear what are the roles of the two losses. It would be more helpful to use examples to show the intent of the two losses. Same for equation 8 and 9. Third, Proposition 1 uses vague descriptions, which is not valid for a mathematical statement.

[1] Liu, Weitang, et al. "Energy-based out-of-distribution detection." Advances in neural information processing systems 33 (2020): 21464-21475.

[2] Sun, Yiyou, et al. "Out-of-distribution detection with deep nearest neighbors." International Conference on Machine Learning. PMLR, 2022.

**Questions:**

1. What's the intuition behind the unknown substitute generation formula (equations 8 and 9)? What is the intuition of including $l_2$ and $l_3$ into the learning objective? Can you use examples to explain them?
2. In line 127, the second kind of substitute is generated from nodes with low classification confidence. Would that also be affected by the over-confident problem?
3. In equation 14, how the $\lambda$'s are selected?
4. I suggest moving line 155-160 (the introduction of GNN) to the background section.

**Limitations:**

Yes

---

> ### Author Rebuttal · Authors · 2024-08-07
>
> **W1:** An important claim made …
>
> **A-W1:** We support the claim through three propositions. The defined propositions can be found in Section 3, and the proofs can be found in Appendix C. In proposition 1, we proved loss $l_1$ and $l_2$ would result in lower energy scores for the model's outputs on training data, i.e. IND data, by taking the partial derivatives of the model parameters $\theta$ with respect to these losses. In Proposition 2, we prove that the overall energy scores of known class samples decrease during the energy propagation, while the overall energy scores of substitute nodes increase, implicitly enlarging the decision boundary between IND and OOD samples. In proposition 3, we prove that loss $l_3$, i.e. the energy regularization loss, reduces the energy scores for IND data and increase the energy scores of OOD data, having the same effect as $l_1$ and $l_2$，and leading to a decrease in the energy scores of IND data. With these three propositions, the proposed method can guarantee an overall distinguishable margin between the detection scores for IND and OOD samples. Thank you. We will add more analysis and explanation in the revision, to improve the clarity.
>
> **W2:** The experiments…
>
> **AW2-1:**  We have conducted comprehensive experiments, including near open-set classification with one unknown class (Table 1 and 7) or multiple unknown classes (Table 6), and far open-set classification with multiple unknown classes (Table 3). We also discussed the experiments of different settings in terms of inductive learning (Table 1 and 6) and transductive learning (Table 7). In addition, we also conducted ablation study (Table 2), parameter analysis (Figure 1\&2), and backbone architectures (Table 8).
>
>  **AW2-2:**  For the papers given here ([1][2]), the main task of these articles is OOD detection, which is a different task with open-set node classification. The goal of OOD detection is to identify the unknown class samples, which is a binary-class classification problem, i.e., a test sample is an OOD sample or not. However, open-set classification requires to identify unknown class samples and classify the known class samples. It is a multiple-class classification problem, which is more difficult then OOD detection. Thus, OOD detection methods cannot be directly applied to open set classification. And this is the main reason we did not include OOD detection methods as baselines. In addition, the proposed method does not reduce IND accuracy much. As shown in Table 5, the average performance across five datasets shows that, compared to the closed-world method GCN, the IND classification accuracy of our method only decreases 3.2\% ( from 79.5\% to 76.6\%), while the accuracy in OOD has been improved from 0\% to 76.1\%.
>
> **AW2-3**: Thanks. Many classic articles [3][4] in this task do not include error bars, and we followed their format. But we agree that error bars can provide more detailed information, and we will add them in the revision.
>
> [3] M. Wu, et al. Openwgl: open-world graph learning for unseen class node classification. Knownledge and Information Systems, pages 1-26 2021.
> [4] L. Shu, et al. Doc:Deep open classification of text document. Empirical Methods in Natural Language Processing, pages 2911-2916, 2017.
>
> **AW2-4** : Thanks. We have conducted this ablation study, as shown in Table 2, i.e. the first two versions without loss $l_3$. Comparing the results of the $l_1$+$l_2$ and $l_1$+$l_2$+$l_3$, we can see that the energy propagation module contributes to the improvement of the model performance, which demonstrates its significance of as an important component.
>
> **W3:** The presentation…
>
> **AW3-1:**  We will address the format issues, and carefully proofread the whole article to improve the overall presentation.
>
> **AW3-2:** The intuition of $l_2$ is related to data generation process. It ensures that the known class samples are sufficiently close to their own class center in the output space.
>
> The intuition of $l_3$ is to control the energy values of known class samples being low, while keeping the energy values of substitute nodes being high, thereby allowing the energy model to remain in an optimal state where it can function effectively. Thus, the role of the loss function $l_3$ is to adjust the energy scores and keep the energy model in an optimal state.
>
> The intuition of Eq. 8 is to generate substitute node at the boundary between two different known classes. For example, we can draw a line between a node from class A and a node from class B, then take a node near the center of this line as constructed inter-class unknown substitute.
>
> The intuition of Eq. 9 is to generate substitute nodes located at the periphery of known classes. The goal is to enhance the model’s ability to recognize unknown classes located at the periphery. We achieve this by selecting class centers of known classes and peripheral nodes, with semantic and structural criteria, then mapping them to generate substitute nodes located at the peripheral of known classes.
>
> **AW3-3:** We will revise it and make it more concise and consistent with mathematical statement. Thanks,
>
> **Q1:** What's the intuition behind…
>
> **A1:** Please refer to **AW3-2** .
>
> **Q2:** In line 127, the second kind of substitute…
>
> **A2:** Thanks. Overconfidence has very slight effect on the generation of our second kind of substitute nodes, since we only need a part of unconfident nodes to help the generation, and we do not require all the unconfident nodes. Thus, there is redundant space to allow the existence of nodes which should be unconfident but are misclassified with overconfidence.
>
> **Q3:** In equation 14…
>
> **A3:** We used a grid search to determine the values of these two hyperparameters. The results are given in E.9 in the Appendix.
>
> **Q4:** line 155-160…
>
> **A4:** We will revise it. Thanks.

---

> > ### Comment · Reviewer_exuD · 2024-08-10
> >
> > Thank you for your detailed response to the concerns raised. I appreciate the effort you've put into addressing each point.
> >
> > After reviewing your explanations, I will increase my score, as many of my concerns have been addressed. However, I hope the presentation can be improved further to enhance the clarity of your work.

---

> > > ### Author Response · Authors · 2024-08-13
> > >
> > > Thank you so much for your valuable comments and suggestions on our manuscript. We appreciate your expertise and careful consideration.  And we will improve the presentation further and enhance the clarity of our work. Thanks again.

---

### Official Review · Reviewer_ena4 · 2024-07-08

**Soundness:** 3
**Presentation:** 3
**Contribution:** 2
**Rating:** 7
**Confidence:** 4

**Summary:**

This paper proposed a new energy-based generative open-set node classification method to achieve open-set graph learning. It uses energy-based models to estimate the underlying density of the seen classes and to evaluate the probability of a given input belonging to the IND classes or OOD classes. Besides, it mimics the distribution of real open-set classes by generating substitute unknown samples under the guidance of graph structure. Besides, the proposed method has nice theoretical properties that guarantee an overall distinguishable margin between the detection scores for IND and OOD samples.

**Strengths:**

1)This paper studies a significant and interesting problem, and the method can be used in a wide range of real-world applications.
2)The paper is overall well motivated. The proposed model is reasonable and sound. Theoretical analysis is performed.
3)Comprehensive experiments, including near open-set classification, far open-set classification, ablation studies and parameter learning, are conducted. The results demonstrate good performance.

**Weaknesses:**

1) What is difference between open-set node classification and out-of-distribution detection problem? The authors should illustrate the differences and whether the proposed method can solve these two problems simultaneously.
2) In section 3.1, it claims that the insouciantly selected unrelated data does not help the open-set classification, thus substitute unknown nodes near the class boundaries are generated. Why randomly selected data does not help? Aren’t they real-world open-set data which can help the open-set learning?
3) It is a good idea to imitating data from open-set distribution by generating pseudo out-of-distribution samples, however the diversity of the generated data is essential. The method used in the paper is manifold mixup, does this method can ensure the diversity of the generated data?

**Questions:**

See Weaknesses part.

**Limitations:**

See Weaknesses part.

---

> ### Author Rebuttal · Authors · 2024-08-07
>
> **Review 3:**
>
> **Weakness 1:** What is difference between open-set node classification and out-of-distribution detection problem? The authors should illustrate the differences and whether the proposed method can solve these two problems simultaneously.
>
> **Answer W1:** Thanks for your suggestion. We will take this into account and emphasise it in the next version.
>  OOD detection is to identify the unknown samples that the model did not learn during the training, which is a binary-class classification problem, i.e., a test sample is an OOD sample or not. Open-set classification requires to identify unknown class samples and classify the known class samples, i.e. it is a multiple-class classification problem, which is more difficult than OOD detection problem. Thus, OOD detection methods cannot be directly applied to the open set classification scenarios. We will add these content in the revision.
>
> **Weakness 2:** In Section 3.1, it claims that the insouciantly selected unrelated data does not help the open-set classification, thus substitute unknown nodes near the class boundaries are generated. Why randomly selected data does not help? Aren't they real-world open-set data which can help the open-set learning?
>
> **Answer W2:**  Thanks for your question. Randomly selected, unrelated data typically does not contain feature information relevant to known classes, providing no boundary confidence to help the model establish classification boundaries. This effectively introduces noise into the open-set classification task, interfering with the model's learning process. In contrast, the unknown class nodes constructed near the class boundaries can provide rich boundary information for establishing classification boundaries, thereby helping the model to understand and recognise potential unknown class categories, improving the model's robustness and classification performance. Although these randomly selected unrelated data may belong to open-set data in the real world, they do not always contain information we need to support open-set learning. In particular, open-set learning relies more on samples that can provide boundary information to help the model determine decision boundaries and narrow the decision space.
>
> **Weakness 3:** It is a good idea to imitating data from open-set distribution by generating pseudo out-of-distribution samples, however the diversity of the generated data is essential. The method used in paper is manifold mixup, does this method can ensure the diversity of the generated data?
>
> **Answer W3:** Thanks for your question. Diversity of generated data is critical when creating pseudo-unknown class points. The construction of clear and unambiguous decision boundaries depends on the high quality and diversity of the generated data. Manifold Mixup is a method that performs data mixing in manifold space, aiming to generate new pseudo-samples by linearly mixing samples. While Manifold Mixup can increase data diversity, it also depends on the mixing strategy used and the characteristics of the data itself. In future work, we will consider the possibility of combining Manifold Mixup with other methods, such as data augmentation and adversarial sample generation, to further ensure the diversity of the generated data.

---

### Official Review · Reviewer_1Tms · 2024-07-09

**Soundness:** 3
**Presentation:** 3
**Contribution:** 3
**Rating:** 7
**Confidence:** 4

**Summary:**

This paper focuses on energy-based open-set node classification, and adopted a generative method to obtain the explicit specific score of a node belonging to the ‘unknown‘ class. To generate good substitute unknowns, it adopted energy-based models to estimate the density of classes and guarantee the nice theoretical properties of the proposed method with an overall distinguishable margin between the detection scores from IND and OOD samples. Extensive experiments are conducted and show superior performance.

**Strengths:**

(1)	The paper is well-structured and overall easy to follow.

(2)	The proposed method can obtain an explicit specific score of belonging to OOD data for each input, which provide more information and show the confidence of the model on its decision.

(3)	The proposed method has nice theoretical properties that guarantee an overall distinguishable margin between the detection scores for IND and OOD samples, the adopted energy regularization loss has consistent effects with the cross-entropy loss as well as with the tailored complement entropy loss on the known classes, and that they are not mutually exclusive.

(4)	The proposed method is agnostic to specific GNN architecture, which demonstrate its generalization for some content.

**Weaknesses:**

(1)	In the main part of the paper, some important experiment settings are missing, which makes the readers feel confused about the motivations of the experiments.

(2)	What are the differences between near open-set classification and far open-set classification? Why do they matter? If a method can achieve good far open-set classification, it should also be good at near open-set classification?

(3)	For section 3, the model design is quite complex, the illustration should be more straightward and make each term and each symbol has enough explanation.

(4)	Some sentences should be improved to make them clearer, such as line 271-273: “Inspired by the Elastic Network Zou and Hastie [2005], Friedman et al. [2010], and considering the similarity in form and function between regularization terms and corresponding error terms…”

**Questions:**

See above.

**Limitations:**

Yes

---

> ### Author Rebuttal · Authors · 2024-08-07
>
> **Review 2:**
>
> **Weakness 1:** In the main part of the paper, some important experiment settings are missing, which makes the readers feel confused about the motivations of the experiments.
>
> **Answer W1:** Thanks for your comment. A more detailed  experimental setting can be found in **subsection E.4** in **Appendix**. Due to space limitations, we have simplified and shortened this part in the main context. We will move the important details from Appendix in the revision.
>
> **Weakness 2:** What are the differences between near open-set classification and far open-set classification? why do they matter? If a method can achieve good far open-set classification, it should also be good at near open-set classification?
>
> **Answer W2:** Thanks for your question. Near open-set classification is to identify the fine-grained OOD samples while also classify the known class samples. Far open-set classification is to identify the coarse-grained OOD samples and also classify the known class samples. In near open-set classification, near ood data may be similar to some of the known classes, thus it requires the model to discern between these easily confusing  categories, and learn very clear and specific boundaries for known classes. In comparison, far ood data normally significantly differ  from known classes. It matters since recognizing categories that are significantly different from the training data is essential to ensuring model robustness. In some content,  we think near open-set classification is more challenging since it requires learning very precise class boundaries to distinguish between fine-grained differences in categories.
>
> **Weakness 3:** For section 3, the model design is quite complex, the illustration should be more straightward and make each term and each symbol has enough explanation.
>
> **Answer W3:** Thanks for your suggestion. We will improve the illustration of **Section 3** and make each term and symbol explained more clearly.
>
> **Weakness 4:** Some sentences should be improved to make them clearer, such as line 271-273:"Inspired by the Elastic Network Zou and Hastie[3005], Friedman et al.[2010], and considering the similarity in form and function between regularization terms and corresponding error terms..."
>
> **Answer W4:** Thanks for your suggestion. We will revise these sentences and improve the whole paper in terms of writing.

---

> > ### Comment · Reviewer_1Tms · 2024-08-11
> >
> > Thank you for the response, which solves most of my concerns, and I will maintain my score on this paper.

---

> > > ### Author Response · Authors · 2024-08-13
> > >
> > > Thank you so much for your valuable comments and suggestions on our manuscript. We appreciate your expertise and careful consideration. We will improve the manuscript further as you suggested. Thanks again.

---

### Official Review · Reviewer_qzuy · 2024-07-10

**Soundness:** 3
**Presentation:** 2
**Contribution:** 3
**Rating:** 5
**Confidence:** 4

**Summary:**

In this paper, the authors proposes a new generative open set node classification method (EGonc) to address the challenge of Open Set Classification (OSC) for safely deploying machine learning models in an open world. Traditional softmax-based methods are overly confident on unseen data, making them vulnerable to out-of-distribution (OOD) samples. EGonc uses graph structure to generate substitute unknowns simulating real open set samples and employs Energy-Based Models (EBM) for density estimation. The method learns additional energy logits from feature residuals to indicate OOD-ness, ensuring a distinguishable margin between in-distribution (IND) and OOD detection scores. Experimental results demonstrate the superiority of EGonc.

**Strengths:**

1. Novel Approach: The paper introduces a new method (EGonc) for open set classification that leverages energy-based models, which is a significant departure from traditional softmax-based methods.
2. Theoretical Guarantees: EGonc comes with strong theoretical properties that ensure a distinguishable margin between in-distribution and out-of-distribution samples.
3. Robustness to OOD Variability: By simulating real open set samples and learning virtual OOD classes, the method enhances robustness against the diversity of OOD examples.

**Weaknesses:**

1. The motivation primarily focuses on the limitations of softmax-based neural networks without delving deeply into the broader implications and potential impact of improved open set classification across various domains and real-world applications.
2. The advantages of the proposed method could be strengthened by providing more concrete examples of practical scenarios where current OSC methods fail and how EGonc specifically addresses these failures, making a stronger case for its real-world applicability and importance.
3. While the method shows promise, its applicability and performance in domains outside of those tested in the experiments need further validation.

**Questions:**

1. Why were these datasets chosen, and what advantages do they offer in validating the proposed method?
2. Why was the 2017 GCN by Kipf and Welling chosen as the backbone neural network for experimental evaluation, instead of using other models?

**Limitations:**

See the weaknesses.

---

> ### Author Rebuttal · Authors · 2024-08-07
>
> **Review 1:**
>
> **Weakness 1:** The motivation primarily focuses on the limitations of softmax-based neural networks without delving deeply into the broader implications and potential impact of improved open set classification across various domains and real-world application.
>
> **Answer W1:** Thanks for your comment. A majority of neural network based classification models use softmax to achieve classification and that is why we mainly focus on softmax-based neural networks. As shown in experiments, in various fields and practical applications, our method can improve the adaptability and generalization of the models in terms of open-set classification scenarios, compared to softmax-based neural networks for closed-world classification, thereby broadening their application scope. In addition, in **Table 1** of the main text, we also include sigmoid-based neural networks (GCN\_sig and GCN\_sig\_$\tau$) as baselines.
> We will add these analysis into the revision.
>
> **Weakness 2:** The advantages of the proposed method could be strengthened by providing more concrete examples of practical scenarios where current OSC methods fail and how EGonc specifically addresses these failures, making a stronger case for its real-world applicability and importance.
>
> **Answer W2:** Thank you for your suggestion. We will specifically present the bad cases of EGonc, $\mathcal{G}^2pxy$ and GNNSAFE across five datasets in the revision, and we will visualise their open set node classification results in the Appendix, make the differences between the three models more intuitively.
>
> **Weakness 3 \& Question 1:** While the method shows promise, its applicability and performance in domains outside of those tested in the experiments need further validation. Why were those datasets chosen, and what advantages do they offer in validating the proposed method?
>
> **Answer W3 \& Q1:** Thanks for your comment and your question. In the field of open-set node classification on graphs, these datasets are the most commonly used. For example, [1] conducted experiments using Cora, Citeseer, Dblp, and Pubmed; our experiments mainly followed their setup and additionally included the obgn\_arxiv dataset to evaluate the model's performance on large-scale data. Specifically, these datasets have the following advantages: First, these datasets (Cora, Citeseer, Dblp, Pubmed, and Ogbn\_arxiv) are widely used in the academic community, are publicly available and accessible, and have a high degree of representativeness and ubiquity, which allows for a fair and equitable measurement of model performance and enhances the transparency and credibility of research. Second, the datasets come from a variety of sources, including computer science (Citeseer, Dblp), medicine (Pubmed), and arxiv preprints (Ogbn\_arxiv), which allows us to validate the applicability and robustness of the proposed methodology in different domains. Third, they have various dataset sizes, including small datasets (Cora, Citeseer), medium datasets (Dblp, Pubmed) and large datasets (Ogbn\_arxiv), which allow us to evaluate the performance of the models with different dataset sizes.
>
> [1]: Man Wu, Shirui Pan, and Xingquan Zhu. Openwgl: open-world graph learning for unseen class node classification. Knownledge and Information Systems, pages 1-26 2021.
>
> **Question 2:** Why was the 2017 GCN by Kipf and Welling chosen as the backbone neural network for experimental evaluation, instead of using other models?
>
> **Answer 2:** Thanks for your question. Actually, the proposed method, EGonc, is agnostic to specific GNN architectures and demonstrates robust generalization capabilities. We conduct experiments on different GNN architectures, including GCN, GAT and GraphSage. The results are shown in **Table 8** in **Appendix E.10**, which have confirm the effectiveness and generalization ability of EGonc for open-set node classification.
> Considering the space limitation, we only illustrate the experimental results in terms of GCN-based EGonc in the body of the article, since GCN is one of the most widely used and one of the most representative GNNs.

---

> > ### Comment · Reviewer_qzuy · 2024-08-09
> >
> > Thank you for your detailed response. I will keep my positive rating.

---

> > > ### Author Response · Authors · 2024-08-13
> > >
> > > Thank you so much for your valuable comments and suggestions on our paper. We appreciate your expertise and careful consideration. And if we have addressed your major concerns, we would like to kindly request if you could increase your score of our paper. We will improve the manuscript further as you suggested. Thanks so much.

---

### Decision · Program_Chairs · 2024-09-25

**Decision:**

Accept (poster)

**Comment:**

This paper proposes a new energy-based open-set node classification method for open-set learning tasks. Initially, the paper received borderline scores. During the rebuttal phase, the authors successfully addressed most concerns, leading to a consensus to accept the paper.

Please follow the reviewers' suggestions to integrate the rebuttal content into the original paper.